# STOCHASTIC OPTIMIZATION WITH NON-STATIONARY NOISE: THE POWER OF MOMENT ESTIMATION

## ABSTRACT

We investigate stochastic optimization under weaker assumptions on the distribution of noise than those used in usual analysis. Our assumptions are motivated by empirical observations in training neural networks. In particular, standard results on optimal convergence rates for stochastic optimization assume either there exists a uniform bound on the moments of the gradient noise, or that the noise decays as the algorithm progresses. These assumptions do not match the empirical behavior of optimization algorithms used in neural network training where the noise level in stochastic gradients could even increase with time. We address this nonstationary behavior of noise by analyzing convergence rates of stochastic gradient methods subject to changing second moment (or variance) of the stochastic oracle. When the noise variation is known, we show that it is always beneficial to adapt the step-size and exploit the noise variability. When the noise statistics are unknown, we obtain similar improvements by developing an online estimator of the noise level, thereby recovering close variants of RMSProp (Tieleman and Hinton, 2012). Consequently, our results reveal why adaptive step size methods can outperform SGD, while still enjoying theoretical guarantees.

## 1 INTRODUCTION

Stochastic gradient descent (SGD) is one of the most popular optimization methods in machine learning because of its computational efficiency compared to traditional full gradient methods. Great progress has been made in understanding the performance of SGD under different smoothness and convexity conditions (Agarwal et al., 2009; Arjevani et al., 2019; Drori and Shamir, 2019; Ghadimi and Lan, 2012; 2013; Nemirovsky and Yudin, 1983; Rakhlin et al., 2012). These results show that with a fixed step size, SGD can achieve the minimax optimal convergence rate for both convex and nonconvex optimization problems, provided the gradient noise is uniformly bounded.

Yet, despite the theoretical minimax optimality of SGD, adaptive gradient methods (Duchi et al., 2011; Kingma and Ba, 2014; Tieleman and Hinton, 2012) have become the methods of choice for training deep neural networks, and have received a surge of attention recently (Agarwal et al., 2018; Chen et al., 2019; Huang et al., 2019; Levy, 2017; Levy et al., 2018; Li and Orabona, 2019; Liu et al., 2019; 2020; Ma and Yarats, 2019; Staib et al., 2019; Ward et al., 2019; Zhang et al., 2019; 2020; Zhou et al., 2018; 2019; Zou and Shen, 2018; Zou et al., 2019). Instead of using fixed stepsizes, these methods construct their stepsizes adaptively using the current and past gradients. Despite advances in the literature on adaptivity, theoretical understanding of the benefits of adaptation is very limited.

We provide a different perspective on understanding the benefits of adaptivity by considering it in the context of non-stationary gradient noise, i.e., the noise intensity varies with iterations. Surprisingly, this setting is rarely studied, even for SGD. To our knowledge, this paper is the first work to formally study stochastic gradient methods in this varying noise scenario. Our main goal is to show that:

*Adaptive step-sizes can guarantee faster rates than SGD when the noise is non-stationary.*

We focus on this goal based on several empirical observations (Section 2), which lead us to model the noise of stochastic gradient oracles via the following iteration dependent quantities:

$$m_k^2 := \mathbb{E}[\|g(x_k)\|^2], \qquad \text{or} \qquad \sigma_k^2 := \mathbb{E}[\|g(x_k) - \nabla f(x_k)\|^2], \qquad (1)$$

where $g(x_k)$ is the stochastic gradient and $\nabla f(x_k)$ the true gradient at iteration $k$. Notation (1) provides more fine-grained description of noise behavior than uniform bounds on the variance by

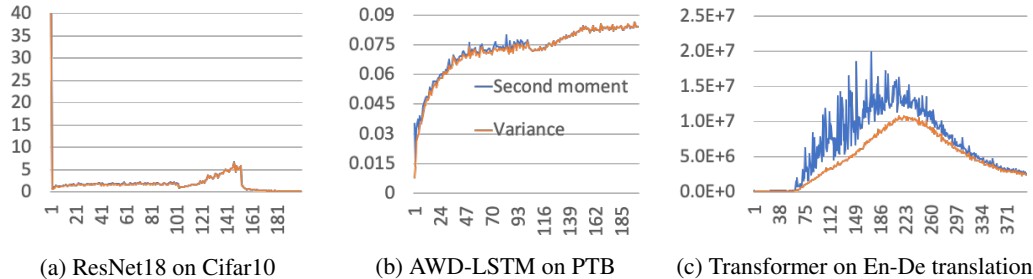

Figure 1: We empirically evaluate the second moment (in blue) and variance (in orange) of stochastic gradients during the training of neural networks. We observe that the magnitude of these quantities changes significantly as iteration count increases, ranging from 10 times (ResNet) to $10^6$ times (Transformer). This phenomenon motivates us to consider a setting with non-stationary noise.

permitting iteration dependent noise intensities. It is intuitive that one should prefer smaller stepsizes when the noise is large and vice versa. Thus, under non-stationarity, an ideal algorithm should adapt its stepsize according to the parameters $m_k$ or $\sigma_k$, suggesting a potential benefit of adaptive stepsizes.

**Contributions.** The primary contribution of our paper is to show that a stochastic optimization method with adaptive stepsize can achieve a faster rate of convergence (by a factor that is polynomial-in-$T$) than fixed-step SGD. We first analyze an idealized setting where the noise intensities are known, using it to illustrate how to select noise dependent stepsizes that are provably more effective (Theorem 1). Next, we study the case with *unknown* noise, where we show under an appropriate smoothness assumption on the noise variation that a variant of RMSProp (Tieleman and Hinton, 2012) can achieve the idealized convergence rate (Theorem 3). Remarkably, this variant does not require the noise levels. Finally, we generalize our results to nonconvex settings (Theorems 12 and 14).

## 2 MOTIVATING OBSERVATION: NONSTATIONARY NOISE IN DEEP LEARNING

Neural network training involves optimizing an empirical risk minimization problem of the form $\min_x f(x) := \frac{1}{n} \sum_{i=1}^n f_i(x)$, where each $f_i$ represents the loss function with respect to the $i$-th data or minibatch. Stochastic methods optimize this objective randomly sampling an incremental gradient $\nabla f_i$ at each iteration and using it as an unbiased estimate of the full gradient. The noise intensity of this stochastic gradient is measured by its second moments or variances, defined as,

1. *Second moment*: $m^2(x) = \frac{1}{n} \sum_{i=1}^n \|\nabla f_i(x)\|^2$;

2. *Variance*: $\sigma^2(x) = \frac{1}{n} \sum_{i=1}^n \|\nabla f_i(x) - \nabla f(x)\|^2$, where $\nabla f(x)$ is the full gradient,

To illustrate how these quantities evolve over iterations, we empirically evaluate them on three popular tasks of neural network training: ResNet18 training on Cifar10 dataset for image classification[1], LSTM training on PTB dataset for language modelling[2]; transformer training on WMT16 en-de for language translation[3]. The results are shown in Figure 1, where both the second moments and variances are evaluated using the default training procedure of the original code.

On one hand, the variation of the second moment/variance has a very different shape in each of the considered tasks. In the CIFAR experiment, the noise intensity is quite steady after the first iteration, indicating a fast convergence of the training model. In LSTM training, the noise level increases and converges to a threshold. While, in training Transformers, the noise level increases very fast at the early epochs, then reaches a maximum, and turns down gradually.

On the other hand, the preferred optimization algorithms in these tasks are also different. For CIFAR10, SGD with momentum is the most popular choice. While for language models, adaptive methods such as Adam or RMSProp are the rule of thumb. This discrepancy is usually taken as granted, based on empirical validation; and little theoretical understanding of it exists in the literature.

---

[1]Code source for CIFAR10 https://github.com/kuangliu/pytorch-cifar

[2]Code source for LSTM https://github.com/salesforce/awd-lstm-lm

[3]Code source for Transformer https://github.com/jadore801120/attention-is-all-you-need-pytorch

Based on the observations made in Figure 1, a natural candidate emerges to explain this discrepancy in the choice of algorithms: the performance of stochastic algorithms varies according the characteristics of gradient noise encountered during training. Despite this behavior, noise level modeling has drawn surprisingly limited attention in prior art. Reference (Moulines and Bach, 2011) studies convergence of SGD assuming each component function is convex and smooth; extensions to the variation of the full covariance matrix are in (Gadat and Panloup, 2017). A more fine-grained stochastic oracle assumes that the variances grow with the gradient norm as $\sigma^2 + c\|\nabla f(x)\|^2$, or grow with the suboptimality $\sigma^2 + c\|x - x^*\|^2$ (Bottou et al., 2018; Jofré and Thompson, 2019; Rosasco et al., 2019).

Unfortunately, these existing oracles fail to express the variation of noise observed in Figure 1. Indeed, the norm of the full gradient, represented as the difference between the orange and the blue line, is significantly smaller compared to the noise level. This suggests that noise variation is *not due* to the gradient norm, but due to some implicit properties of the objective function. This observation motivates us to introduce the following non-stationary noise oracle:

**Definition 1** (**non-stationary noise oracle**). *The stochasticity of the problem is governed by a sequence of second moments $\{m_k\}_{k\in\mathbb{N}}$ or variances $\{\sigma_k\}_{k\in\mathbb{N}}$, such that, at the $k_{th}$ iteration, the gradient oracle returns an unbiased gradient $g(x_k)$ such that $\mathbb{E}[g(x_k)] = \nabla f(x_k)$, and either*

*(a)  with second moment $\mathbb{E}[\|g(x_k)\|^2] = m_k^2$; or*

*(b)  with variance $\mathbb{E}[\|g(x_k) - \nabla f(x_k)\|^2] = \sigma_k^2$.*

The non-stationary noise oracle is a relaxation of the standard uniform noise oracle in which case $m_k$ or $\sigma_k$ are constant. By introducing the time dependency, we aim to understand how the variation of noise influences the convergence rate of optimization algorithms. An example that falls into this category is when the noise is additive to the gradient, namely $g(x_k) \sim \nabla f(x_k) + \mathcal{N}(0, \sigma_k^2)$.

We emphasize that our goal is to demystify the correlation between the noise intensity and the performance of optimization algorithm, instead of explaining why certain shape of noise occurs. In general, the variation in noise is a consequence of the combination on data distribution, training model and optimization method, which is complex and highly non trivial. We simplify it by assuming that the noise intensity is decoupled from its location, meaning that, the parameters $m_k$ or $\sigma_k$ only depend on the iteration number $k$, but do not depend on the specific location where the gradient is evaluated. This is empirically justified as the pattern of the noise is mostly determined by the task instead of the optimization algorithms, see Appendix A. The simplification helps to focus on the shape of noise, taking a first step towards the goal: ***characterize the convergence rate of adaptive algorithms under non-stationary noise.***

## 3    THE BENEFIT OF ADAPTIVITY UNDER NONSTATIONARY NOISE

In this section, we investigate the influence of nonstationary noise in an idealized setting where *the noise parameters $m_k$ are known*. For brevity, we will first focus on the convex setting and present our results based on the second moment parameters $m_k$. We defer the discussion on **nonconvex** problems and variants on variance parameters $\sigma_k$ to Section 5. One reason that we prioritize the second moment than the variance is to draw a connection with the well-known adaptive method RMSProp (Tieleman and Hinton, 2012) and Adam (Kingma and Ba, 2014). One common feature shared by both algorithms is that they scale the step sizes inversely to an exponential moving average of estimated second moments. Below, we start from the idealized case assuming the second moments are known and show why inverse scaling could speed up convergence.

Let $f$ be convex and differentiable. We consider the problem $\min_x f(x)$, where the gradient is given by the nonstationary noise oracle defined in Definition 1. We assume that the optimum is attained at $x^*$ and we denote $f^*$ the minimum of the objective. We are interested in studying the convergence rate of a stochastic algorithm with update rule

$$x_{k+1} = x_k - \eta_k g(x_k), \tag{2}$$

where $\eta_k$ are stepsizes that are oblivious of the iterates $\{x_k\}_{k\in\mathbb{N}}$.

**Theorem 1.** *Under the second moment oracle given in Definition 1 (a), the weighted average $\overline{x}_T = (\sum_{k=1}^T \eta_k x_k)/(\sum_{k=1}^T \eta_k)$ obtained by the update rule (2) satisfies the suboptimality bound*

$$\mathbb{E}[f(\overline{x}_T) - f^*] \leq \frac{\|x_1 - x^*\|^2 + \sum_{k=1}^T \eta_k^2 m_k^2}{\sum_{k=1}^T \eta_k}. \tag{3}$$

Although the theorem follows from standard analysis, it leads to valuable results as explained below.

**Corollary 2.** *Denote $M = \max m_k$ and $R = \|x_1 - x^*\|$. We have the following convergence rate:*

1. *SGD with constant stepsize: if $\eta_k = \eta = \frac{R}{\sqrt{\sum_{k=1}^{T} m_k^2}}$, then*

$$\mathbb{E}[f(\overline{x}_T) - f^*] \leq \frac{2R\sqrt{\sum_{k=1}^{T} m_k^2}}{T} = \frac{2RM}{\sqrt{T}} \cdot \sqrt{\frac{\frac{1}{T}\sum_{k=1}^{T} m_k^2}{M^2}}. \qquad \text{(constant baseline)}$$

2. *SGD with idealized stepsize: if $\eta_k = \frac{R}{\sqrt{T}m_k}$, then*

$$\mathbb{E}[f(\overline{x}_T) - f^*] \leq \frac{2R\sqrt{T}}{\sum_{k=1}^{T} \frac{1}{m_k}} = \frac{2RM}{\sqrt{T}} \cdot \frac{\frac{1}{M}}{\frac{1}{T}\sum_{k=1}^{T} \frac{1}{m_k}} \cdot \qquad \text{(idealized baseline)}$$

To facilitate comparison, we have normalized the convergence rates with respect to the conventional rate $2RM/\sqrt{T}$ (Nemirovski et al., 2009). In the standard setting, the values of $m_k$ are unavailable, but the uniform bound $M$ is known, in such a case taking $m_k = M$ recovers the standard result.

When the values of $m_k$ are given, both *constant baseline* and *idealized baseline* benefits from it, improving upon the conventional rate $2RM/\sqrt{T}$. The improvement factor in *constant baseline* has depends on the average of the second moments $\sqrt{\sum m_k^2/T}$ , while as the improvement factor in *idealized baseline* depends on the average of the harmonic sum $\frac{1}{T}\sum 1/m_k$. In particular, from Jensen's inequality $\mathbb{E}[X]^{-2} \leq \mathbb{E}[X^{-2}]$, we have

$$(\tfrac{1}{T}\textstyle\sum_k \tfrac{1}{m_k})^{-2} \leq \tfrac{1}{T}\textstyle\sum_k m_k^2,$$

implying that the *idealized baseline* is always better than the *constant baseline*. This result is rather expected, as the stepsizes are adapted to the noise intensity.

As a consequence, the accumulations of the parameters $m_k$ in different forms governed the convergence rate. To further illustrate such difference, we consider an illustrative synthetic noise model, mimicking the shape of noise we observed in the training of Transformer (see Figure 1(c)).

**Example 1.** *Consider the following piece-wise linear noise model with $\gamma = 5(1 - T^{-\alpha})/T$.*

$$m_k = \begin{cases} \frac{1}{T^\alpha} & \text{if } k \in [1, \frac{T}{5}]; \\ \gamma(k - \frac{2T}{5}) + 1 & \text{if } k \in (\frac{T}{5}, \frac{2T}{5}]; \\ 1 & \text{if } k \in (\frac{2T}{5}, \frac{3T}{5}]; \\ \gamma(\frac{3T}{5} - k) + 1 & \text{if } k \in (\frac{3T}{5}, \frac{4T}{5}]; \\ \frac{1}{T^\alpha} & \text{if } k \in (\frac{4T}{5}, T]. \end{cases}$$

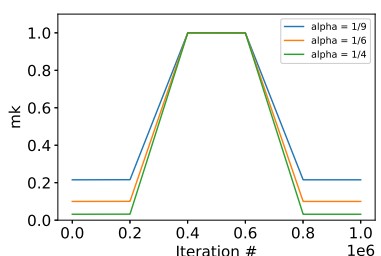

In this example, the maximum noise intensity $M$ is 1, and the minimum intensity is $1/T^\alpha$, inducing a large ratio of order $T^\alpha$. Following the bounds developed in Corollary 2, the performance of the *constant baseline* maintains the standard $O(1/\sqrt{T})$ convergence rate, while as the *idealized baseline* converges at $O(1/T^{\frac{1}{2}+\alpha})$. Hence a nontrivial acceleration of order $T^\alpha$ is obtained by using the idealized stepsize, and this acceleration can be arbitrarily large as $\alpha$ increases.

This example is encouraging, showing that the speedup due to adaptive stepsizes can be polynomial in the number of iterations, especially when the ratio between the maximum and the minimum noise intensity is large. However, explicit knowledge on $m_k$ is required to implement these idealized stepsizes, which is unrealistic. The next sections of the paper demonstrates that estimating the moment bound in an online fashion can achieve a convergence rate comparable to the idealized setting.

## 4 ADAPTIVE METHODS: ONLINE ESTIMATION OF MOMENTS

From now on, we assume that the moment bounds $m_k$ are not given. To address the non-stationarity, we estimate the noise intensity based on an exponential moving average, a technique commonly used in adaptive methods. More precisely, the moment estimator $\hat{m}_k$ is constructed recursively as

$$\hat{m}_{k+1}^2 = \beta\hat{m}_k^2 + (1 - \beta)\|g_k\|^2, \qquad \text{(ExpMvAvg)}$$

---

**Algorithm 1** Adaptive SGD $(x_1, T, c, m)$

1: Initialize $\hat{m}_1^2 = \|g(x_1)\|^2$.
2: **for** $k = 1, 2, ..., T$ **do**
3:     Evaluate stochastic gradient $g_k$ at $x_k$.
4:     $x_{k+1} = x_k - \eta_k g_k$ with $\eta_k = \frac{c}{\hat{m}_k + m}$.
5:     $\hat{m}_{k+1}^2 = \beta \hat{m}_k^2 + (1 - \beta)\|g_k\|^2$.
6: **end for**
7: **return** $\overline{x}_T = (\sum_{i=1}^T \eta_i x_i)/(\sum_{i=1}^T \eta_i)$.

---

where $g_k$ is the $k$-th stochastic gradient and $\beta$ is the decay paramter. Then we choose the stepsizes inversely proportional to $\hat{m}_{k+1}$, leading to Algorithm 1.

Algorithm 1 could be viewed as a "norm" version of RMSProp (Tieleman and Hinton, 2012): the exponential moving average is performed coordinate-wise in RMSProp, whereas we use the full norm of $g_k$ to update the moment estimator $\hat{m}_{k+1}$. Such a simplification via a full norm variant has also been analyzed in the uniformly bounded noise setting (Levy, 2017; Levy et al., 2018; Li and Orabona, 2019; Ward et al., 2019)—we leave investigation of the more advanced coordinate-wise version as a topic of future research.

Another important component of the stepsize is the correction constant $m$, the term appearing in the denominator. This constant provides a safety threshold when $\hat{m}_k$ underestimates $m_k$, which is commonly used in the practical implementation of adaptive methods, and even beyond, in reinforcement learning as so-called exploration bonus (Azar et al., 2017; Jin et al., 2018a; Strehl and Littman, 2008).

To show the convergence of the algorithm, we need to impose a regularity assumption on the sequence of noise intensities. Otherwise, previous estimate may not provide any information of the next one.

**Assumption 1.** *We assume that an upper bound $M$ on $m_k$ is given, i.e. $\max_k m_k \leq M$ such that*

*(a) The fourth moment of $g_k$ is bounded by $M^4$, namely, $\mathbb{E}[(\|g_k\|^2 - m_k^2)^2] \leq M^4, \forall k$.*

*(b) The total variation on $m_k$ is bounded, namely $\sum_k |m_k^2 - m_{k+1}^2| \leq D^2$ with $D^2 = \Omega(M^2)$.*

The bounded fourth moment ensures concentration of $\|g_k\|^2$, which is necessary to guarantee the quality of the online estimator. In particular, this assumption is satisfied when $\|g_k\|$ follows a $m_k$ **sub-Gaussian** distribution. Though stronger than bounded variance, this assumption *does not* lead to better convergence rates for SGD, as many existing results (including lower bounds) require sub-Gaussian or bounded noise, see Agarwal et al. (2009); Ghadimi and Lan (2013) .

The bounded total variation assumption should be viewed as a regularity condition on the sequence of noise. It is motivated and common in the dynamic online learning literature (Besbes et al., 2014; Jadbabaie et al., 2015; Mokhtari et al., 2016). A key aspect of it is to avoid infinite oscillation, such as the pathological setting where $m_{2k} = 1$ and $m_{2k+1} = M$, in which case the total variation scales with the number of iterations $T$. The specific constant in $D^2 = \Omega(M^2)$ depends on the shape of the noise. When $m_k$ is increasing in the first half and decreasing in the second half, as in the Transformer experiments and Example 1, the total variation is bounded by $D^2 \leq 2M^2$. More generally, if the noise can be decomposed into $K$ piece-wise monotone fragments, then the bound $D^2 \leq KM^2$ holds.

With the above assumptions, we are now ready to present our convergence result.

**Theorem 3.** *Under Assumptions 1, with probability at least $1/2$, the iterates generated by Algorithm 1 using parameters $\beta = 1 - 2T^{-2/3}$, $m = 4\sqrt{D^2 + M^2}T^{-\frac{1}{9}}\ln(T)^{\frac{1}{2}}$, $c = \frac{R}{\sqrt{T}}$ satisfy*

$$f(\overline{x}_T) - f^* \leq \frac{2RM}{\sqrt{T}} \cdot \frac{\frac{32}{M}}{\frac{1}{T}\sum_{k=1}^T \frac{1}{m_k + m}}.$$

**Remark 4.** *Our result directly implies a $1 - \delta$ high probability style convergence rate, by restarting it $2\log(1/\delta)$ times. An additional $\log(1/\delta)$ dependency will be introduced in the complexity, as in standard high probability results Fang et al. (2018); Jin et al. (2018b); Nemirovski et al. (2009).*

The key to prove the theorem is to effectively bound the estimation error $|\hat{m}_k^2 - m_k^2|$ relying on concentration, and on bounded variation in Assumption 1. In particular, the choice of the decay

| | Constant | Adaptive | Idealized |
|---|---|---|---|
| $0 \leq \alpha \leq \frac{1}{9}$ | $\mathcal{O}\left(T^{-\frac{1}{2}}\right)$ | $\tilde{\mathcal{O}}\left(T^{-\frac{1+2\alpha}{2}}\right)$ | $\mathcal{O}\left(T^{-\frac{1+2\alpha}{2}}\right)$ |
| $\frac{1}{9} < \alpha$ | $\mathcal{O}\left(T^{-\frac{1}{2}}\right)$ | $\tilde{\mathcal{O}}\left(T^{-\frac{11}{18}}\right)$ | $\mathcal{O}\left(T^{-\frac{1+2\alpha}{2}}\right)$ |

Table 1: Comparison of the convergence rate under the noise example 1.

parameter $\beta$ is critical, determining how fast the contribution of past gradients decays. Because of the non-stationarity in noise, the online estimator $\hat{m}_k$ is biased. The proposed choice of $\beta$ carefully balances the bias error and the variance error, leading to sublinear regret, see Appendix C.

Due to the correction constant $m$, the obtained convergence rate inversely depends on $\sum_{k=1}^{T} \frac{1}{m_k+m}$, instead of the idealized dependency $\sum_{k=1}^{T} \frac{1}{m_k}$. This additional term makes the comparison less straightforward and we now discuss different scenarios for obtaining a better understanding.

### 4.1 DISCUSSION OF THE CONVERGENCE RATE

To illustrate the difference between convergence rates, we first consider the synthetic noise model introduced in Example 1. The detailed comparison is presented in Table 1, where we observe two regimes regarding the exponent $\alpha$:

- When $0 \leq \alpha \leq \frac{1}{9}$, the rate of the adaptive algorithm matches (*idealized baseline*) up to logarithmic dependency, and is $T^\alpha$ better than the (*constant baseline*).

- When $\frac{1}{9} \leq \alpha$, the adaptive convergence rate no longer matches the (*idealized baseline*). Nevertheless, it is always $T^{\frac{1}{9}}$ faster than the (*constant baseline*).

In both cases, the adaptive method achieves a non-trivial improvement, polynomial in $T$, compared to the (*constant baseline*). Even though the improvement $T^{\frac{1}{9}}$ might seem in-significant, it is the first result showing a plausible non-trivial advantage of adaptive methods over SGD under nonstationary-noise. Further, note that the adaptive convergence rate does not always match the (*idealized baseline*) when $\alpha$ is large. Such a discrepancy comes from the correction term $m$, which makes the stepsize more conservative than it should be, especially when $m_k$ is small.

The above comparison relies on the specific choice on the noise model given in Example 1. Now we formalize some simple conditions allowing comparison in more general settings.

**Corollary 5.** *If the ratio $M/(\min_k m_k) \leq T^{\frac{1}{9}}$, then adaptive method converges in the same order as the (idealized baseline), up to logarithmic dependency.*

This result is remarkable since the adaptive method does not require any knowledge of $m_k$ values, and yet it achieves the idealized rate. In other words, the exponential moving average estimator successfully adapts to the variation in noise, allowing faster convergence than constant stepsizes.

**Corollary 6.** *Let $m_{avg}^2 = \sum m_k^2/T$ be the average second moment. If $M/m_{avg} \leq T^{\frac{1}{9}}$, then adaptive method is no slower than the (constant baseline), up to logarithmic dependency.*

The condition in Corollary 6 is strictly weaker than the condition in Corollary 5, which means even though an adaptive method may not match the idealized baseline, it could still be non-trivially better than the constant baseline. This case happens e.g., when $\alpha > \frac{1}{9}$ in Table 1, where the adaptive method is $\mathcal{O}(T^{\frac{1}{9}})$ faster than the constant baseline. Indeed, $\mathcal{O}(T^{\frac{1}{9}})$ is the maximum improvement one can expect according to our current analysis.

**Corollary 7.** *Recall that $M$ is an upper bound on $m_k$, i.e., $\max m_k \leq M$. Therefore*

1. *The convergence rate of the constant baseline is no slower than $\mathcal{O}(2RM/\sqrt{T})$.*

2. *The convergence rate of the adaptive method is no faster than $\tilde{\mathcal{O}}(2RM/T^{\frac{1}{2}+\frac{1}{9}})$.*

The order of maximum improvement $\mathcal{O}(T^{\frac{1}{9}})$ is determined by the specific choice of $m$ in Theorem 3, which is chosen to be $\tilde{\mathcal{O}}(MT^{-\frac{1}{9}})$. Indeed, the correction term is helpful when the estimator $\hat{m}_k$ underestimates the true value $m_k$, avoiding the singularity at zero. Hence, the choice of $m$ is related

to the average deviation between $\hat{m}_k$ and $m_k$. Under a stronger concentration assumption, we can strengthen the maximum improvement to $\mathcal{O}(T^{\frac{1}{6}})$, as shown in Appendix E.

The noise model in Example 1 provides a favorable scenario where the maximum improvement is attained. However, in some scenarios, the convergence rate of an adaptive method can be slower than the constant baseline.

**Adversarial scenario.** If $m_k = 1/T^\alpha$ for all $i \in [1, T]$ except at $T/2$ it takes the value $m_{T/2} = 1$ with $\alpha > 1/9$, then the convergence rate of both constant and idealized baselines are $O(T^{-\frac{1}{2}})$, while the adaptive method only converges in $\tilde{\mathcal{O}}(T^{-\frac{1-2\alpha}{2}})$. The subtle change at iteration $T/2$ amplifies the exponential moving average estimator and requires a non-negligible period to get back to the constant level. It is clear that the estimator becomes less meaningful under such a subtle change.

Overall, it is hard to provide a complete characterization of the variation in noise. In Corollary 5 and 6, we show that when the ratio between the maximum and the minimum/average second moment is not growing too fast, adaptive methods do improve upon SGD.

## 5   EXTENSIONS OF THM 3

In this section, we discuss several extensions to Thm 3. The results are nontrivial but the analysis is almost the same. Hence we defer the exact statements and proofs to appendices.

**Addressing the variance oracle.** So far, we have focused on the noise oracle based on the second moment $m_k$ and made the connection with existing adaptive methods. However, there is some unnaturalness underlying the non-stationary oracle on $m_k$. Indeed, it is hard to argue that $m_k$ is iterate independent since $m_k^2 = \sigma_k^2 + \|\nabla f(x_k)\|^2$. Even though the influence of $\|\nabla f(x_k)\|^2$ might be minor when the variance $\sigma_k^2$ is high (e.g. as in Figure 1), it is still changing $m_k$. In contrast, the variance $\sigma_k$ is an intrinsic quantity coming from the noise model, which could be iterate independent. Hence the variance oracle is theoretically more sound. We now present the necessary modifications in order to adapt to the variance oracle.

First, in order to estimate the variance, we need to query two stochastic gradients $g_k$ and $g'_k$ at the same iterate, then we construct the estimator following the recursion

$$\hat{\sigma}_{k+1}^2 = \beta\hat{\sigma}_k^2 + (1 - \beta)\|g_k - g'_k\|^2.$$

Second, the smoothness condition on $f$ is required, i.e., $L$-Lipschitzness on the gradient of $f$. In this case, it is necessary to ensure that the step-size being not larger than $1/2L$. This translates to an additional constraint on the correcting constant $m$. More precisely, the stepsize is given by

$$\eta_k = \frac{c}{\hat{\sigma}_k + m} \quad \text{with} \quad m \geq 2cL.$$

Note that the $L$-smoothness condition is not required in the second moment oracle. This is why the second moment oracle is more suitable to nonsmooth setting (see Section 6.1 of Bubeck (2014)). A complete algorithm for the variance oracle is provided in Algorithm 2. The convergence results are essentially the same by replacing $m_k$ with $\sigma_k$, see Appendix H.

**Extension to nonconvex setting.** We also provide an extension of our analysis to non-convex smooth setting. In which case, we characterize the convergence with respect to the gradient norm $\|\nabla f(x_k)\|^2$, i.e., convergence to a stationary point. The conclusions are very similar to the one in the convex setting and the results (Theorems 12 and 14) are deferred to Appendix F.

**Variants on stepsizes.** To go beyond the second moment of noise, one could apply an estimator of the form $\hat{m}_{k+1}^p = \beta\hat{m}_k^p + (1 - \beta)\|g_k\|^p$ when the $p$-th moment of the gradient is bounded. This allows stepsize of the shape $\eta_k \propto 1/(\hat{m}_k^p + m^p)^{1/p}$ as in ADAM, Adamax Kingma and Ba (2014).

## 6   EXPERIMENTS

In this section, we describe two sets of experiments that verify the faster convergence of Algorithm 1 against vanilla SGD. The first experiment is on linear regression with synthetic noise described in Example 1 and the second set of experiments is on neural network training.

### 6.1   SYNTHETIC EXPERIMENTS

In the synthetic experiment, we generate a random linear regression dataset using the sklearn[4] library. We design the stochastic oracle as full gradient with injected Gaussian noise, whose coordinate-wise

---

[4]scikit-learn.org/stable/modules/generated/sklearn.datasets.make_regression.html

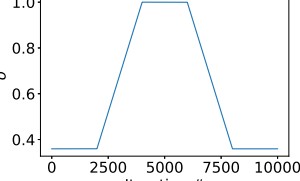 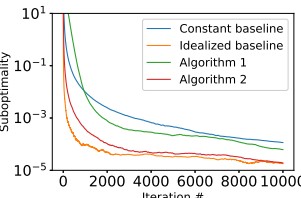 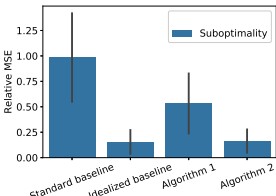

Figure 2: **Left:** The injected noise intensity over iterations. **Middle:** Average loss trajectory over 10 runs for four different algorithms: standard baseline, idealized baseline, Alg 1 and Alg 2. The curve (idealized vs standard) confirms that adopting step sizes inverse to the noise level lead to faster convergence and less variations. **Right:** Average and standard deviation of function suboptimality. The values are normalized by the average MSE of the standard baseline.

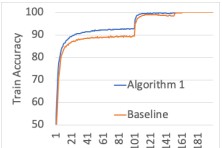 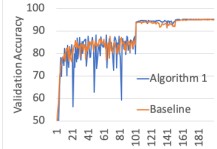 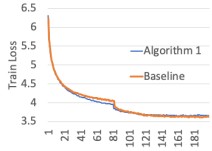 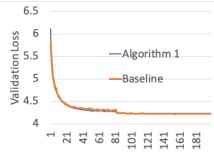

Figure 3: The left two plots show the accuracy of training ResNet18 on Cifar10 dataset. The right two plots present the negative log-likelihood loss for LSTM Language modelling from Merity et al. (2018). The baselines are provided by the repos cited on page 2. Algorithm 1 is described in Alg 1.

standard deviation $\sigma$ is shown in the left figure of Fig 2. We then run the four algorithms discussed in this work: standard baseline, idealized baseline, Alg 1 and Alg 2. We finetune the step sizes for each algorithm by grid-searching among $10^k$, where $k$ is an integer. We repeat the experiment for 10 runs and show the average training trajectory as well as the function suboptimality in Fig 2. We observe that the performance is ranked as follows: idealized baseline, Alg 2, Alg 1 and standard baseline.

### 6.2 NEURAL NETWORK TRAINING

We demonstrate how the proposed algorithm performs in real-world neural network training. We first tested our algorithm on Cifar10 classification task. We then implement our algorithm into the AWD-LSTM codebase described in Merity et al. (2018). We see from Figure 3 that our proposed algorithm can achieve slightly better performance than baselines. Despite our main contribution being providing theoretical analysis for the fast convergence of adaptive methods with moment estimation, these results show that our analysis can also lead to efficient and practical algorithm design.

Besides convergence, we also measured noise level during neural network training with different optimizers and found that the noise pattern is mostly determined by the learning task instead of by the optimization algorithm. Details can be found in Appendix A.

## 7 CONCLUSIONS

This paper discusses convergence rates of stochastic gradient methods in an empirically motivated setting where the noise level is changing over iterations. We show that under mild assumptions, one can achieve faster convergence than the fixed step SGD by a factor that is polynomial in number of iterations, by applying online noise estimation and using adaptive step sizes. Our analysis, therefore provides one explanation for the recent success of adaptive methods in neural network training.

There is much more to be done along the line of non-stationary stochastic optimization. Under our current analysis, there is a gap between the adaptive method and the idealized method when the noise variation is large (see second row in Table 1). A natural question to ask is whether one could reduce this gap, or alternatively, is there any threshold preventing the adaptive method from getting arbitrarily close to the idealized baseline? Moreover, could one attain further acceleration by combining momentum or coordinate-wise update techniques? Answering these questions would provide more insight and lead to a better understanding of widely used adaptive methods.

Perhaps a more fundamental question is regarding the iterate dependency: the setting where the moments $m_k$ or the variance $\sigma_k$ are functions of the current update $x_k$, not just of the iteration index $k$. Significant effort needs to be spent to address this additional correlation under appropriate regularity conditions. We believe our work lays the foundation to address this challenging research problem.

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

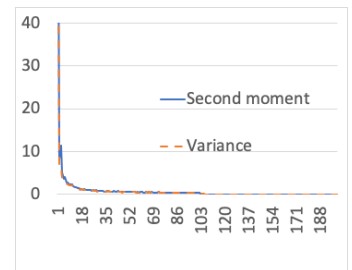 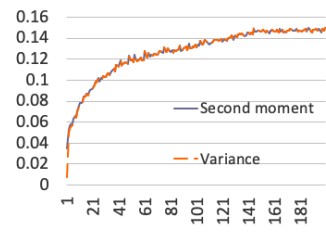

(a) ResNet18 trained with Adam on Cifar10     (b) AWD-LSTM trained with Adam on PTB

Figure 4: We retrain the models with Adam optimizer and evaluate the second moment (in blue) and variance (in orange) of stochastic gradients for the Cifar10 experiments and PTB experiments in Figure 1.

## A    ADDITIONAL EXPERIMENT DETAILS AND RESULTS

In this section, we provide more details on our experiment setup. The code for reproducing noise estimation and neural network training on AWS-LSTM model is uploaded in the supplementary. The code for Cifar10 classification can be uploaded upon request.

### A.1    DETAILS ON PTB TRAINING

Our implementation is based on the author's github repository [5]. The original codebase trains the network using clipped gradient descent followed by an average SGD (ASGD) algorithm to prevent overfitting. As generalization error is beyond our discussion, we focus on the first phase (which takes about 200 epochs) by removing the ASGD training part. Aside of the number of epochs, all other parameters are the same as the default training procedure. For our algorithm, we update the clipping value to be $0.25$, and set the hyperparameters of Algorithm 1 as $\eta_k = 5, \beta = 0.95, m = 0.01$.

### A.2    DETAILS ON CIFAR10 TRAINING

Our implementation is based on a pytorch implementation [6] of the original ResNet paper(He et al., 2016). We train the ResNet18 model on Cifar10 dataset for 200 epochs. The baseline uses SGD with learning rate initialized as $0.1$. The learning rate is decayed by 10 at epochs $100, 150$. We eventually achieves approximately $95\%$ validation accuracy. For our algorithm, we set $\eta_k = 1$, $\beta = 0.99, m = 0.01$ and use the same learning rate schedule as the baseline.

### A.3    NOISE BEHAVIOR FOR DIFFERENT ALGORITHMS

In this subsection, we include one interesting observation on the noise pattern of neural network training. From Figure 1, we see that the noise pattern looks very different for different task. One may naturally wonder whether the difference results from the particular training task or from the optimization algorithms. Noticing that each of these three tasks are trained with a different algorithm (Cifar10 with momentum SGD; PTB with clipped SGD; En-De translation with ADAM), we retrained the Cifar10 task and the PTB language modelling with Adam. and rerun the En-De translation experiment with momentum SGD. We found that the En-De translation task diverges when trained with SGD even when learning rate is $0.001$. Hence we only report the results from the other two experiments in Figure 4.

From Figure 4, we see that the pattern looks very similar to the original plots in Figure 1. Hence these experiments suggest that the noise pattern is mostly determined by the learning task instead of by the optimization algorithms.

---

[5]Code source for LSTM https://github.com/salesforce/awd-lstm-lm
[6]Code source for CIFAR10 https://github.com/kuangliu/pytorch-cifar

## B  PROOF OF THEOREM 1

*Proof.* The iterate suboptimality have the following relationship:
$$\|x_{k+1} - x^*\|^2 = \|x_k - \eta_k g_k - x^*\|^2 = \|x_k - x^*\|^2 - 2\eta_k \langle g_k, x_k - x^* \rangle + \eta_k^2 \|g_k\|^2.$$
Rearrange and take expectation with respect to $g_k$ we have
$$2\eta_k(f(x_k) - f^*) \leq 2\eta_k \langle \nabla f(x_k), x_k - x^* \rangle$$
$$\leq \mathbb{E}\|x_{k+1} - x^*\|^2 - \mathbb{E}\|x_k - x^*\|^2 + \eta_k^2 m_k^2$$
Sum over $k$ and take expectation we get
$$\mathbb{E}[\sum_{k=1}^{T} 2\eta_k(f(x_k) - f^*)] \leq \|x_1 - x^*\|^2 + \sum_{k=1}^{T} \eta_k^2 m_k^2$$
Then from convexity, we have
$$\mathbb{E}[f(\overline{x}_T) - f^*] \leq \frac{\|x_1 - x^*\|^2 + \sum_{k=1}^{T} \eta_k^2 m_k^2}{\sum_{k=1}^{T} \eta_k},$$
where $\overline{x}_T = (\sum_{i=1}^{T} \eta_i x_i)/(\sum_{i=1}^{T} \eta_i)$. Corollary 2 follows from specifying the particular choices of the stepsizes.

$\square$

## C  KEY LEMMA

**Lemma 8.** *Under Assumptions 1, taking $\beta = 1 - 2T^{-2/3}$, the total estimation error of the $\hat{m}_k^2$ based on* (ExpMvAvg) *is bounded by:*
$$\mathbb{E}\left[\sum_{k=1}^{T} |\hat{m}_k^2 - m_k^2|\right] \leq 2(D^2 + M^2)T^{2/3}\ln(T^{2/3})$$

*Proof.* On a high level, we decouple the error in a bias term and a variance term. We use the total variation assumption to bound the bias term, and use the exponential moving average to reduce variance. Then we pick $\beta$ to balance the two terms.

From triangle inequality, we have
$$\sum_{k=0}^{T} \mathbb{E}\left[|\hat{m}_k^2 - m_k^2|\right] \leq \sum_{k=1}^{T} \underbrace{\mathbb{E}\left[|\hat{m}_k^2 - \mathbb{E}[\hat{m}_k^2]|\right]}_{\text{Variance term}} + \sum_{k=1}^{T} \underbrace{\left|\mathbb{E}[\hat{m}_k^2] - m_k^2\right|}_{\text{Bias term}} \quad (4)$$
We first bound the bias term. By definition of $\hat{m}_k$, we have
$$\mathbb{E}[\hat{m}_k^2] - m_k^2 = \beta\mathbb{E}[\hat{m}_{k-1}^2] + (1 - \beta)m_{k-1}^2 - m_k^2$$
$$= \beta(\mathbb{E}[\hat{m}_{k-1}^2] - m_{k-1}^2) + (m_{k-1}^2 - m_k^2)$$
Hence by recursion,
$$\mathbb{E}[\hat{m}_k^2] - m_k^2 = \beta^{k-1}\underbrace{(\mathbb{E}[\hat{m}_1^2] - m_1^2)}_{=0} + \beta^{k-2}(m_1^2 - m_2^2) + \cdots + (m_{k-1}^2 - m_k^2)$$
Therefore, the bias term could be bounded by
$$\sum_{k=1}^{T} \left|\mathbb{E}[\hat{m}_k^2] - m_k^2\right| \leq \sum_{k=1}^{T}\sum_{j=1}^{k-1} \beta^{k-1-j}\left|m_j^2 - m_{j+1}^2\right|$$
$$= \sum_{k=1}^{T-1} \left|m_k^2 - m_{k+1}^2\right| \sum_{j=0}^{T-1-k} \beta^j$$
$$\leq \frac{1}{1 - \beta}\sum_{k=1}^{T-1} \left|m_k^2 - m_{k+1}^2\right|$$
$$\leq \frac{D^2}{1 - \beta} \quad \text{(From Assumption (1))}$$

The first inequality follows by traingle inequality. The third inequality uses the geometric sum over $\beta$. To bound the variance term, we remark that

$$\hat{m}_k^2 = (1-\beta)g_{k-1}^2 + (1-\beta)\beta g_{k-2}^2 + \cdots + (1-\beta)\beta^{k-2}g_1^2 + \beta^{k-1}g_0^2.$$

Hence from independence of the gradients, we have

$$\mathbb{E}\left[|\hat{m}_k^2 - \mathbb{E}[\hat{m}_k^2]|\right] \leq \sqrt{\text{Var}[\hat{m}_k^2]}$$

$$= \sqrt{\text{Var}[(1-\beta)g_{k-1}^2] + \text{Var}[(1-\beta)\beta g_{k-2}^2] + \cdots + \text{Var}[(1-\beta)\beta^{k-2}g_1^2] + \text{Var}[\beta^{k-1}g_0^2]}$$

$$\leq \sqrt{(1-\beta)^2 + (1-\beta)^2\beta^2 + \cdots + (1-\beta)^2\beta^{2(k-2)} + \beta^{2(k-1)}}M^2,$$

where $M^2$ is an upperbound on the variance. The first inequality follows by Jensen's inequality. The second equality uses independence of $g_i$ given $g_1, ..., g_{i-1}$. The last inequality follows by assumption 1.

We distinguish two cases, when $k$ is small, we simply bound the coefficient by 1, i.e.

$$\sqrt{(1-\beta)^2 + (1-\beta)^2\beta^2 + \cdots + (1-\beta)^2\beta^{2(k-2)} + \beta^{2(k-1)}} \leq 1$$

When $k$ is large such that $k \geq 1+\gamma$, with $\gamma = \frac{1}{2(1-\beta)}\ln(\frac{1}{1-\beta})$, we have $\beta^{2(k-1)} \leq 1-\beta$, thus

$$\sqrt{(1-\beta)^2 + (1-\beta)^2\beta^2 + \cdots + (1-\beta)^2\beta^{2(k-2)} + \beta^{2(k-1)}}$$

$$\leq \sqrt{\frac{(1-\beta)^2}{1-\beta^2} + \beta^{2(k-1)}}$$

$$\leq \sqrt{\frac{(1-\beta)^2}{1-\beta^2} + (1-\beta)}$$

$$\leq \sqrt{2(1-\beta)}$$

The second inequality follows by $k \geq 1+\gamma$, with $\gamma = \frac{1}{2(1-\beta)}\ln(\frac{1}{1-\beta})$. Therefore, when $k \geq 1+\gamma$,

$$\mathbb{E}\left[|\hat{m}_k^2 - \mathbb{E}[\hat{m}_k^2]|\right] \leq \sqrt{2(1-\beta)}M$$

Therefore, substitute in the above equation into the

$$\sum_{k=1}^{T}\mathbb{E}\left[|\hat{m}_k^2 - \mathbb{E}[\hat{m}_k^2]|\right] = \sum_{k=1}^{\gamma}\mathbb{E}\left[|\hat{m}_k^2 - \mathbb{E}[\hat{m}_k^2]|\right] + \sum_{k=\gamma+1}^{T}\mathbb{E}\left[|\hat{m}_k^2 - \mathbb{E}[\hat{m}_k^2]|\right]$$

$$\leq (\gamma + (T-\gamma)\sqrt{2(1-\beta)})M^2$$

Summing up the variance term and the bias term yields,

$$\sum_{k=0}^{T}\mathbb{E}\left[|\hat{m}_k^2 - m_k^2|\right] \leq \frac{D^2}{1-\beta} + (\gamma + (T-\gamma)\sqrt{2(1-\beta)})M^2 \tag{5}$$

Taking $\beta = 1 - T^{-2/3}/2$ yields,

$$\sum_{k=0}^{T}\mathbb{E}\left[|\hat{m}_k^2 - m_k^2|\right] \leq 2(D^2 + M^2)T^{2/3}\ln(T^{2/3}) \tag{6}$$

$\square$

# D    PROOF OF THEOREM 3

On a high level, the difference between the adaptive stepsize and the idealized stepsize mainly depends on the estimation error $|\hat{m}_k^2 - m_k^2|$, which has a sublinear regret according to Lemma C. Then we carefully integrate this regret bound to control the derivation from the idealized algorithm, reaching the conclusion.

*Proof.* By the update rule of $x_{k+1}$, we have,

$$\|x_{k+1} - x^*\|^2 = \|x_k - \eta_k g_k - x^*\|^2 = \|x_k - x^*\|^2 - 2\eta_k \langle g_k, x_k - x^* \rangle + \eta_k^2 \|g_k\|^2.$$

Noting that the stepsize $\eta_k$ is independent of $g_k$, taking expectation with respect to $g_k$ conditional on the past iterates lead to

$$\begin{aligned}
2\eta_k(f(x_k) - f^*) &\leq 2\eta_k \langle \nabla f(x_k), x_k - x^* \rangle \\
&= \mathbb{E}[2\eta_k \langle g_k, x_k - x^* \rangle | x_k, \cdots, x_1] \\
&= -\mathbb{E}[\|x_{k+1} - x^*\|^2 | x_k, \cdots, x_1] + \|x_k - x^*\|^2 + \eta_k^2 m_k^2.
\end{aligned}$$

Recall that $R = \|x_1 - x^*\|$, taking expectation and sum over iterations $k$, we get

$$\mathbb{E}[2(\sum_{k=1}^{T} \eta_k)(f(\overline{x}_T) - f^*)] \leq R^2 + \mathbb{E}[\sum_{k=1}^{T} \eta_k^2 m_k^2].$$

Hence by Markov's inequality, with probability at least $3/4$,

$$2(\sum_{k=1}^{T} \eta_k)(f(\overline{x}_T) - f^*) \leq 4\mathbb{E}[2(\sum_{k=1}^{T} \eta_k)(f(\overline{x}_T) - f^*)] \leq 4(R^2 + \mathbb{E}[\sum_{k=1}^{T} \eta_k^2 m_k^2]). \quad (7)$$

Now we can upper bound the right hand side, indeed

$$\begin{aligned}
\sum_{k=1}^{T} \mathbb{E}[\eta_k^2 m_k^2] &= c^2 \sum_{k=1}^{T} \mathbb{E}\left[\frac{m_k^2}{(\hat{m}_k + m)^2}\right] \\
&\leq c^2 \left(\sum_{k=1}^{T} \mathbb{E}\left[\frac{m_k^2 - \hat{m}_k^2}{(\hat{m}_k + m)^2}\right] + \sum_{k=1}^{T} \mathbb{E}\left[\frac{\hat{m}_k^2}{(\hat{m}_k + m)^2}\right]\right) \\
&\leq c^2 \left(\frac{1}{m^2} \sum_{k=1}^{T} \mathbb{E}\left[|m_k^2 - \hat{m}_k^2|\right] + T\right) \\
&\leq c^2 \left(\frac{(M^2 + D^2)T^{2/3} \ln(T^{2/3})}{m^2} + T\right) \leq 3c^2 T
\end{aligned} \quad (8)$$

The last inequality follows by the choice on $m$. Hence, from Eq. (7), we have with probability at least $3/4$,

$$2(\sum_{k=1}^{T} \eta_k)(f(\overline{x}_T) - f^*) \leq 4(R^2 + 3c^2 T) \quad (9)$$

Next, by denoting $(x)_+ = \max(x, 0)$, we lower bound the left hand side,

$$\begin{aligned}
\frac{1}{c} \sum \eta_k &= \sum \frac{1}{\hat{m}_k + m} \\
&= \sum \frac{1}{m_k + m} + \sum \left(\frac{1}{\hat{m}_k + m} - \frac{1}{m_k + m}\right) \\
&\geq \sum \frac{1}{m_k + m} - \sum \frac{(\hat{m}_k - m_k)_+}{(m_k + m)(\hat{m}_k + m)} \\
&\geq \sum \frac{1}{m_k + m} - \sum \frac{(\hat{m}_k - m_k)_+}{\sqrt{m_k + m} \cdot m^{3/2}} \\
&\geq \sum \frac{1}{m_k + m} - \sum \frac{1}{2}\left(\frac{(\hat{m}_k - m_k)_+^2}{m^3} + \frac{1}{m_k + m}\right) \\
&= \frac{1}{2} \sum \frac{1}{m_k + m} - \frac{1}{2m^3} \sum (\hat{m}_k - m_k)_+^2
\end{aligned} \quad (10)$$

Finally, by Markov's inequality, with probability $3/4$

$$\sum (\hat{m}_k - m_k)_+^2 \le 4\mathbb{E}[\sum (\hat{m}_k - m_k)_+^2] \le 4\mathbb{E}[\sum (\hat{m}_k - m_k)^2] \le 8(D^2 + M^2)T^{2/3}\ln(T^{2/3}).$$

Following the choice of $m = 4\sqrt{D^2 + M^2}T^{-\frac{1}{9}}\ln(T)^{\frac{1}{2}}$, we have

$$\frac{1}{2m^3}\sum_{k=1}^{T}(\hat{m}_k - m_k)_+^2 \le \frac{T}{4(M+m)} \le \frac{1}{4}\sum_{k=1}^{T}\frac{1}{m_k + m}$$

Consequently, together with (9) and (10), we know that with probability at least $1 - \frac{1}{4} - \frac{1}{4} = 1/2$,

$$f(\overline{x}_T) - f^* \le \frac{4(R^2 + 3c^2 T)}{\sum_k \frac{c}{4(m_k+m)}} \le \frac{2R}{\sqrt{T}} \cdot \frac{32T}{\sum_k \frac{1}{(m_k+m)}}, \tag{11}$$

where the last inequality follows by setting $c = \frac{R}{\sqrt{T}}$.

$\square$

**Remark 9.** *For more general choices of stepsize $\eta_k = \frac{1}{(\hat{m}_k^p + m^p)^{1/p}}$, the upper bound in Eq.(8) holds exactly as in the above proof, and the lower bound in Eq.(10) follows from*

$$\sum \eta_k = \sum \frac{1}{(\hat{m}_k^p + m^p)^{1/p}}$$

$$= \sum \frac{1}{(m_k^p + m^p)^{1/p}} + \sum \left(\frac{1}{(\hat{m}_k^p + m^p)^{1/p}} - \frac{1}{(m_k^p + m^p)^{1/p}}\right)$$

$$= \sum \frac{1}{(m_k^p + m^p)^{1/p}} + \sum \frac{(m_k^p + m^p)^{1/p} - (\hat{m}_k^p + m^p)^{1/p}}{(\hat{m}_k^p + m^p)^{1/p}(m_k^p + m^p)^{1/p}}$$

$$\ge \sum \frac{1}{(m_k^p + m^p)^{1/p}} - \sum \frac{(\hat{m}_k - m_k)_+}{(\hat{m}_k^p + m^p)^{1/p}(m_k^p + m^p)^{1/p}} \quad \textit{(Minkowski inequality)}$$

$$\ge \sum \frac{1}{(m_k^p + m^p)^{1/p}} - \sum \frac{(\hat{m}_k - m_k)_+}{m^{3/2}(m_k^p + m^p)^{1/2p}}$$

$$\ge \frac{1}{2}\sum \frac{1}{(m_k^p + m^p)^{1/p}} - \frac{1}{2m^3}\sum (\hat{m}_k - m_k)_+^2.$$

## E  PROOF WITH CONCENTRATE NOISE

In this section, we add additional constraints on noise concentrations.

**Assumption 2.** *The expected absolute value is not very different from the square root of the second moment, i.e.*

$$2\mathbb{E}[\|g(x)\|] \ge \sqrt{\mathbb{E}[\|g(x)\|^2]}.$$

The constant "2" in the assumption above is arbitrary and can be increased to any fixed constant. The above assumption is satisfied if $g(x)$ follows Gaussian distribution. It is also satisfied if for some fixed constant $\gamma$, $p(\|g(x)\| \ge r) \le \gamma\mathbb{E}[\|g(x)\|]^3 r^{-4}$, for all $r \ge \gamma\mathbb{E}[\|g(x)\|]$.

We assume that the total variation on the first moment is bounded.

**Assumption 3.** *We denote $\lambda_k = \mathbb{E}[\|g_k\|]$, and assume that an upper bound $M$ such that*

(a) *The second moment of $g_k$ is bounded by $M^2$, namely, $\mathbb{E}[\|g_k\|^2] \le M^2, \forall k$.*

(b) *The total variation on the first moment $\lambda_k$ is bounded by*

$$\sum_k |\lambda_k - \lambda_{k+1}| \le D = \Omega(M). \tag{12}$$

| | Constant | Adaptive | Idealized |
|---|---|---|---|
| $0 \leq \alpha \leq \frac{1}{6}$ | $\mathcal{O}\left(T^{-\frac{1}{2}}\right)$ | $\tilde{\mathcal{O}}\left(T^{-\frac{1+2\alpha}{2}}\right)$ | $\mathcal{O}\left(T^{-\frac{1+2\alpha}{2}}\right)$ |
| $\frac{1}{6} < \alpha$ | $\mathcal{O}\left(T^{-\frac{1}{2}}\right)$ | $\tilde{\mathcal{O}}\left(T^{-\frac{2}{3}}\right)$ | $\mathcal{O}\left(T^{-\frac{1+2\alpha}{2}}\right)$ |

Table 2: Comparison of the convergence rate under the noise example 1.

Under this stronger assumption, we can perform our online estimator on the first moment $\mathbb{E}[\|g_k\|]$ instead of the second moment, replacing line 6 of Algorithm 1 by

$$\hat{m}_{k+1} = \beta \hat{m}_k + (1-\beta)\|g_k\|. \tag{13}$$

The theorem below shows the convergence rate of the new algorithm.

**Theorem 10.** *Under assumptions 2, 3, with $m = 16(D+M)T^{-1/6}\ln(T)$, $c = \frac{R}{\sqrt{T}}$, Algorithm 1 with update rule (13) achieves convergence rate*

$$f(\overline{x}_T) - f^* \leq \frac{2R}{\sqrt{T}} \cdot \frac{12T}{\sum_k \frac{1}{(m_k+m)}}, \tag{14}$$

With a better concentration of the online estimator, we could allow a less conservative correction constant $m$, in the order of $MT^{-\frac{1}{6}}$. It is this parameter that controls the maximum attainable improvement compared to the constant baseline. Indeed, we again consider the noise example 1, given in Table 2. In this case, the adaptive method can obtain an improvement of order $T^{\frac{1}{6}}$ compared to the constant baseline, while as previously only $T^{\frac{1}{9}}$ is achievable.

The proof of the result follows a similar routine as the proof of Theorem 3. We start by presenting an equivalent lemma of Lemma 8.

**Lemma 11.** *Under assumption 3, we can achieve the following bound on total estimation error.*

$$\mathbb{E}[\sum_{k=1}^{T} |\hat{m}_k - \lambda_k|] \leq 2(D+M)T^{2/3}\ln(T^{2/3})$$

*Proof.* The proof is the same as the proof of Lemma 8, by replacing the second $m_k^2$ by the first moment $\lambda_k = \mathbb{E}[\|g_k\|]$.

$\square$

*Proof of Theorem 10.* By Assumption 2, we can use first moment of $g_k$ to bound the second moment. Hence, Eq. (7) implies that with probability at least $3/4$,

$$2(\sum_{k=1}^{T} \eta_k)(f(\overline{x}_T) - f^*) \leq 4(R^2 + \mathbb{E}[4\sum_{k=1}^{T} \eta_k^2 \lambda_k^2]). \tag{15}$$

Now we upper bound the right hand side, indeed

$$\sum_{k=1}^{T} \mathbb{E}[\eta_k^2 \lambda_k^2] = c^2 \sum_{k=1}^{T} \mathbb{E}\left[\frac{\lambda_k^2}{(\hat{m}_k+m)^2}\right]$$

$$\leq c^2 \left(\sum_{k=1}^{T} \mathbb{E}\left[\frac{\lambda_k^2 - \hat{m}_k^2}{(\hat{m}_k+m)^2}\right] + \sum_{k=1}^{T} \mathbb{E}\left[\frac{\hat{m}_k^2}{(\hat{m}_k+m)^2}\right]\right)$$

$$\leq c^2 \left(\sum_{k=1}^{T} \mathbb{E}\left[\frac{(\lambda_k - \hat{m}_k)(\lambda_k + \hat{m}_k)}{(\hat{m}_k+m)^2}\right] + T\right)$$

$$\leq c^2 \left(\frac{2M}{m^2}\sum_{k=1}^{T} \mathbb{E}\left[|\lambda_k - \hat{m}_k|\right] + T\right)$$

$$\leq c^2 \left(\frac{4(M+D)MT^{2/3}\ln(T^{2/3})}{m^2} + T\right) \leq 2c^2 T$$

Hence by Markov inequality, with probability at least $3/4$,

$$2(\sum_{k=0}^{T-1} \eta_k)(f(x_I) - f^*) \leq 4\mathbb{E}[2(\sum_{k=0}^{T-1} \eta_k)(f(x_I) - f^*)] \leq 4(R^2 + 2c^2 T)$$

Next, we lower bound the left hand side,

$$\frac{1}{c} \sum \eta_k = \sum \frac{1}{\hat{m}_k + m}$$

$$= \sum \frac{1}{\lambda_k + m} + \sum \left( \frac{1}{\hat{m}_k + m} - \frac{1}{\lambda_k + m} \right)$$

$$\geq \sum \frac{1}{\lambda_k + m} - \sum \frac{(\hat{m}_k - \lambda_k)_+}{(\hat{m}_k + m)(\lambda_k + m)}$$

$$\geq \sum \frac{1}{\lambda_k + m} - \frac{1}{m^2} \sum (\hat{m}_k - \lambda_k)_+$$

By Markov's inequality and Lemma 11, with probability $3/4$, we have

$$\sum (\hat{m}_k - \lambda_k)_+^2 \leq 4\mathbb{E}[\sum |\hat{m}_k - \lambda_k|] \leq 8(D + M)T^{2/3} \ln(T^{2/3}).$$

Following the choice of $m = 16(D + M)T^{-\frac{1}{6}} \ln(T)$, we have

$$\frac{1}{m^2} \mathbb{E}[(\hat{m}_k - \lambda_k)_+] \leq \frac{T}{2(M + m)} \leq \frac{1}{2} \sum \frac{1}{\lambda_k + m}$$

Consequently, we know that with probability at least $1 - \frac{1}{4} - \frac{1}{4} = 1/2$,

$$f(\overline{x}_T) - f^* \leq \frac{4(R^2 + 2c^2 T)}{\sum_k \frac{c}{2(\lambda_k + m)}} \leq \frac{2R}{\sqrt{T}} \cdot \frac{12T}{\sum_k \frac{1}{(m_k + m)}}, \quad (16)$$

by setting $c = \frac{R}{\sqrt{T}}$ and the fact that $\lambda_k \leq m_k$. $\qquad \square$

## F  VARIANCE ORACLE AND EXTENSION TO NONCONVEX SETTING

In this section, we show that we can adapt our algorithm to the variance oracle in Definition 1 (b), where

$$\mathbb{E}[\|g_k - \nabla f(x_k)\|^2] = \sigma_k^2.$$

To avoid redundancy, we present the result in the nonconvex smooth setting. We make the following smoothness assumptions.

**Assumption 4.** *The function is $L-$smooth, i.e. for any $x, y$, $\|\nabla f(x) - \nabla f(y)\| \leq L\|x - y\|$.*

Remark that the $L$-smoothness condition is not required in the second moment oracle. This is why the second moment assumption is usually imposed in the non-differentiable setting (see Section 6.1 of Bubeck (2014)). We first provide the convergences of SGD serving as the baselines.

**Theorem 12** (**Nonconvex baselines**). *Under the variance oracle in Definition 1 (b) and Assumption 4, the convergence of SGD using update $x_{k+1} = x_k - \eta_k g_k$ with $\eta_k \leq \frac{1}{2L}$ satisfies*

$$\mathbb{E}[\|\nabla f(x_I)\|^2] \leq \frac{f(x_1) - f^* + \frac{L}{2} \sum_{k=1}^{T} \eta_k^2 \sigma_k^2}{\sum_{k=1}^{T} \eta_k}, \quad (17)$$

*where $I$ is an random variable with $\mathbb{P}(I = i) \propto \eta_i$.*

This convergence result is very similar to the one shown in the convex setting in Theorem 1. Instead of bounding the function suboptimality, the upper bound is only available on the norm of the gradient, implying convergence to a stationary point. We remark that an additional requirement on the stepsize is required, namely $\eta_k \leq 1/2L$. This is not surprising since taking a large stepsize will not be helpful due to the $L$-smoothness condition. Hence, we can not take a stepsize inversely depending on $\sigma_k$ when the noise is small. This restriction makes the comparison on convergence rate less straightforward. To facilitate the discussion on convergence rate, we make an additional assumption on the lower bound of the variance $\sigma_k$.

---

**Algorithm 2** Variance Adaptive SGD $(x_1, T, c, m)$

---

1: Initialize $\hat{\sigma}_1 = \frac{\|g_1 - g'_1\|^2}{2}$, where $g_1, g'_1$ are two independent stochastic gradients at $x_1$.
2: **for** $k = 1, 2, ..., T$ **do**
3:     Query two independent stochastic gradient $g_k, g'_k$ at $x_k$.
4:     Update $x_{k+1} = x_k - \eta_k(g_k + g'_k)/2$ with $\eta_t = \frac{c}{\hat{\sigma}_k + m}$ and $m \geq 2cL$.
5:     Update $\hat{\sigma}_{k+1}^2 = \beta\hat{\sigma}_k^2 + (1-\beta)\frac{\|g_k - g'_k\|^2}{2}$
6: **end for**
7: **return** $x_I$ where $I$ is the random variable such that $\mathbb{P}(I = i) \propto \eta_i$.

---

**Assumption 5.** *For any $k \in [1, T]$, $\sigma_k \geq \sqrt{8L(f(x_1) - f(x^*))}/\sqrt{T}$.*

We emphasize that the above condition is not necessary to derive convergence analysis, but only for the clarity in terms of comparison on different convergence rate. This assumption help us focus on the case when noise (instead of the shape of the deterministic function) dominates the convergence rate and determines the step size. In otherword, our step size choice under this setting satisfies $\eta_k \leq 1/2L$, leading to the following convergence rate:

**Corollary 13.** *Let $\Delta = f(x_1) - f^*$. We have the following two convergence rate bounds for SGD:*

1. *SGD with constant stepsize: if $\eta_k = \eta = \sqrt{\frac{2\Delta}{L\sum_k \sigma_k^2}}$, then*

$$\mathbb{E}[\|\nabla f(x_I)\|^2] \leq \frac{\sqrt{2L\Delta\sum_{k=1}^{T}\sigma_k^2}}{T} = \sqrt{\frac{2L\Delta}{T}} \cdot \sqrt{\frac{\sum_{k=1}^{T}\sigma_k^2}{T}}. \qquad (constant\ baseline)$$

2. *SGD with idealized stepsize: if $\eta_k = \frac{1}{\sigma_k}\sqrt{\frac{2\Delta}{LT}}$, then*

$$\mathbb{E}[\|\nabla f(x_I)\|^2] \leq \frac{\sqrt{2LT\Delta}}{\sum_{k=1}^{T}\frac{1}{\sigma_k}} = \sqrt{\frac{2L\Delta}{T}} \cdot \frac{T}{\sum_{k=1}^{T}\frac{1}{\sigma_k}}. \qquad (idealized\ baseline)$$

The resulting convergence rate are similar as the convex setting. We now modify the adaptive algorithm to make use of the variance oracle and the smoothness assumption. The through algorithm is presented in Algorithm 2. Particularly, we keep an exponential moving average of the variance instead of the moments, using two stochastic gradients $g_k$ and $g'_k$ at the same iterate,

$$\hat{\sigma}_{k+1}^2 = \beta\hat{\sigma}_k^2 + (1-\beta)\frac{\|g_k - g'_k\|^2}{2}.$$

In particular, $\mathbb{E}[\|g_k - g'_k\|^2/2]$ is an unbiased estimator of $\sigma_k^2$. To provide the convergence analysis of the algorithm, we make the following assumptions on the variation of the variance.

**Assumption 6.** *We assume an upper bound on $\sigma_k^2 = \mathbb{E}[\|g_k - \nabla f(x_k)\|^2]$, i.e. $\max_k \sigma_k \leq M$. We also assume that the total variation in $\sigma_k$ is bounded. i.e. $\sum_k |\sigma_k^2 - \sigma_{k+1}^2| \leq D^2 = 4M^2$.*

With the above assumptions, algorithm 2 achieve the following convergence rate.

**Theorem 14.** *Under the assumptions 4, 6 and $T$ large enough such that $\ln T \leq T^{1/3}$, algorithm 2 with $c = \sqrt{\frac{2\Delta}{LT}}$, $m = 4\sqrt{D^2 + M^2}T^{-1/9}\ln(T)^{1/2} + 2cL$, achieves with probability 1/2,*

$$\|\nabla f(x_I)\|^2 \leq \sqrt{\frac{2L\Delta}{T}} \cdot \frac{32T}{\sum_k \frac{1}{(\sigma_k + m)}}.$$

The above theorem is almost the same as Theorem 3, and hence all the remarks for Theorem 3 also applies in the nonconvex case.

## G    PROOF OF THEOREM 12

*Proof.* By $L$-smoothness, we have

$$f(x_{k+1}) \leq f(x_k) - \eta_k\langle g_k, \nabla f(x_k)\rangle + \frac{L\eta_k^2}{2}\|g_k\|^2$$

Rearrange and take expectation with respect to $g_k$, we get

$$\left(\eta_k - \frac{L\eta_k^2}{2}\right)\|\nabla f(x_k)\|^2 \leq f(x_k) - f(x_{k+1}) + \frac{L\eta_k^2}{2}\mathbb{E}[\|g_k - \nabla f(x_k)\|^2].$$

From the condition $\eta_k \leq \frac{1}{2L}$, we have,

$$\frac{\eta_k}{2}\|\nabla f(x_k)\|^2 \leq f(x_k) - f(x_{k+1}) + \frac{L}{2}\eta_k^2\sigma_k^2.$$

Sum over k and take expectation,

$$\mathbb{E}[\sum_{k=1}^T \eta_k\|\nabla f(x_k)\|^2] \leq f(x_0) - f(x^*) + \frac{L}{2}\sum_k \eta_k^2\sigma_k^2.$$

Denote $I$ as the random variable such that $\mathbb{P}(I = i) \propto \eta_i$. We know

$$(\sum_{k=1}^T \eta_k)\mathbb{E}[\|\nabla f(x_I)\|^2] \leq f(x_0) - f(x^*) + \frac{L}{2}\sum_{k=1}^T \eta_k^2\sigma_k^2.$$

This yields the desired convergence rate in (17). $\qquad\square$

## H    PROOF OF THEOREM 14

The proof is almost identical to the proof of Theorem 3. We start by presenting an equivalent theorem of Lemma 8 below.

**Lemma 15.** *Under assumption 6, we can achieve the following bound on total estimation error using the estimator* (ExpMvAvg):

$$\mathbb{E}[\sum_{k=\gamma}^T |\hat{\sigma}_k^2 - \sigma_k^2|] \leq 2(D^2 + M^2)T^{2/3}\ln(T^{2/3})$$

*Proof.* This follows by exactly the same proof as Theorem 8 and the fact that $\mathbb{E}[\|g - g'\|^2] = 2\mathbb{E}[\|g - \nabla f\|^2]$. $\qquad\square$

*Proof of Theorem 14.* The first part of the proof follows the schema as in the proof of Theorem 12, when $\eta_k \leq 1/2L$, we know that

$$\mathbb{E}[(\sum_{k=1}^T \eta_k)\|\nabla f(x_I)\|^2] \leq \Delta + \frac{L}{2}\sum_k \mathbb{E}[\eta_k^2\sigma_k^2].$$

The rest of the proof is exactly the same as the proof of Theorem 3. We can upper bound the right hand side, indeed

$$\begin{aligned}
\sum_{k=1}^T \mathbb{E}[\eta_k^2\sigma_k^2] &= c^2\sum_{k=1}^T \mathbb{E}\left[\frac{\sigma_k^2}{(\hat{\sigma}_k + m)^2}\right] \\
&= c^2\left(\sum_{k=1}^T \mathbb{E}\left[\frac{\sigma_k^2 - \hat{\sigma}_k^2}{(\hat{\sigma}_k + m)^2}\right] + \sum_{k=1}^T \mathbb{E}\left[\frac{\hat{\sigma}_k^2}{(\hat{\sigma}_k + m)^2}\right]\right) \\
&\leq c^2\left(\frac{1}{m^2}\sum_{k=1}^T \mathbb{E}\left[|\sigma_k^2 - \hat{\sigma}_k^2|\right] + T\right) \\
&\leq c^2\left(\frac{2(M^2 + D^2)T^{2/3}\ln(T^{2/3})}{m^2} + T\right) \leq 3c^2T
\end{aligned}$$

The last inequality follows by the choice of parameters that $\frac{M^2+D^2}{m^2} \leq \frac{1}{16}\frac{T^{1/3}}{\ln(T)}$. Hence by Markov inequality, with probability at least $3/4$,

$$(\textstyle\sum_{k=1}^{T} \eta_k)\|\nabla f(x_I)\|^2 \leq 4\mathbb{E}[(\textstyle\sum_{k=1}^{T} \eta_k)\|\nabla f(x_I)\|^2] \leq 4(\Delta + \frac{3Lc^2T}{2})$$

Next, we lower bound the left hand side as in (10),

$$\frac{1}{c}\sum \eta_k = \sum \frac{1}{\hat{\sigma}_k + m} \geq \frac{1}{2}\sum \frac{1}{\sigma_k + m} - \frac{1}{2m^3}\sum (\hat{\sigma}_k - \sigma_k)_+^2 \qquad (18)$$

Finally, by Markov's inequality, with probability $3/4$,

$$\sum (\hat{\sigma}_k - \sigma_k)_+^2 \leq 4\mathbb{E}[\sum (\hat{\sigma}_k - \sigma_k)_+^2] \leq 4\mathbb{E}[\sum (\hat{\sigma}_k - \sigma_k)^2] \leq 8(D^2 + M^2)T^{2/3}\ln(T^{2/3}).$$

Following the choice of $m = 4\sqrt{D^2 + M^2}T^{-\frac{1}{9}}\ln(T)^{\frac{1}{2}} + 2cL$, we have

$$\frac{1}{2m^3}\sum_{k=1}^{T} (\hat{\sigma}_k - \sigma_k)_+^2 \leq \frac{T}{4(M + m)} \leq \frac{1}{4}\sum_{k=1}^{T} \frac{1}{\sigma_k + m}$$

Together with (18) implies that with probability $3/4$,

$$\sum \eta_k \geq \frac{c}{4}\sum_{k=1}^{T} \frac{1}{\sigma_k + m}$$

Consequently, we know that with probability at least $1 - \frac{1}{4} - \frac{1}{4} = 1/2$,

$$\|\nabla f(x_I)\|^2 \leq \frac{4(\Delta + \frac{3Lc^2T}{2})}{\sum_k \frac{c}{4(\sigma_k+m)}} \leq \sqrt{\frac{2L\Delta}{T}} \cdot \frac{32T}{\sum_k \frac{1}{(\sigma_k+m)}}, \qquad (19)$$

by setting $c = \sqrt{\frac{2\Delta}{LT}}$.

$\square$

