# OpenReview forum: "Stochastic Optimization with Non-stationary Noise: The Power of Moment Estimation"
_ICLR.cc/2021/Conference — Reject_

### Official Review · AnonReviewer4 · 2020-10-28
**New attempt to handle non-stationary gradient noise**

**Rating:** 3
**Confidence:** 5

**Review:**

## Summary

The authors provide a new analysis of SGD and versions of RMSprop, taking into account possible non-stationarity of the gradient noise. In particular, the authors propose.
(i) the convergence analysis of SGD with stepsizes dependent on the second moment of stochastic gradients and a "norm" version of RMSprop (Moment Adaptive SGD) in the convex non-smooth case and
(ii) the convergence analysis of SGD with stepsizes dependent on the variances of stochastic gradients and a variance adaptive SGD in the non-convex smooth case with rates of gradient norm decrease.
The rates are given in terms of high probability convergence.

## Strengths

1) The paper addresses an important and fundamental question for understanding why methods based on moment estimation (like RMSprop or Adam) work better in certain regimes.
2) The experimental study of gradient noise non-stationarity shows interesting behavior of the gradient noise that can be useful for future research in this area.

## Weaknesses

1. The theory presented in the paper needs significant further development and refinement: some results are misleading, assumptions are too strong and not motivated well, several crucial statements are left without proof, and some of the presented proofs have inaccuracies/unexplained parts. Below, I provide a list of my major concerns about the paper.
1.1. **Assumptions.** While authors state in the first sentence of the abstract that they analyze stochastic optimization under "weaker assumptions on the distribution of noise than those used in usual analysis," this is not true. In fact, the main algorithms -- Algorithm 1 and Algorithm 2 -- are analyzed under Assumptions 1 and 6. These assumptions are strictly stronger (otherwise, according to Nemirovski and Yudin lower bounds for stochastic non-smooth optimization, it is impossible to get a better rate than SGD has) than the uniformly bounded second moment of the stochastic gradient and uniformly bounded variance of the stochastic gradient assumptions. Moreover, in the non-smooth convex case, which is considered as the main case of the paper, authors assume that $m_k = \mathbb{E}[||g_k||^2]$, where $g_k$ is an conditionally unbiased estimator of $\nabla f(x_k)$, is bounded and *does not depend on $x_k$*. Moreover, the authors do not explicitly state this as an assumption and do not emphasize this fact in the statements of Theorems 1 and 3. It is an important assumption and should be stated explicitly in the statements of these theorems. Next, $m_k$ depends on $x_k$ even when the noise is negligible, i.e., the stochastic gradients are almost gradients.
  1.2. **Misleading results.** Unfortunately, some results require further explanations. For example, why can we choose $\eta_k$ dependent on $\sum_{t=1}^Tm_t$ in Corollary 2? It is misleading since $m_t$ for $t > k$ depends on $\eta_k$ in general. Next, it is unclear how can we know $m_k$ or even $\sum_{k=1}^T m_k$ to use these stepsizes? It seems, that in general it is not possible.
 1.3. **Results without the proofs.** Some results that play a central role in the paper are left without rigorous proofs and even sketches of the proofs. In particular, the rates from Table 1 are not derived for the noise model considered in Example 1, and the most important results of the paper --- Corollaries 5, 6, and 7 --- are provided without the proofs as well. These results were meant to explain the main advantage of Algorithm 1 in comparison with vanilla SGD, but the proofs for them seem to be not that straightforward and should be added to the paper.
 1.4. **Inaccuracies and unexplained parts in the proofs.** I have checked all proofs in detail. While, in general, they are sound and mathematically correct, some places in the proofs of the main convergence results for Algorithm 1 and 2 are left without explanation and seem to be inaccurate (see my list with comments below).
2. Clarity of the paper should be improved. Some claims require further clarification; see my comments below.
3. The paper contains a significant amount of typos and grammatical errors. Here are some of them:
 - page 4, "The improvement factor ...": has depends $\to$ depends
 - page 13, "The iterate suboptimality ...": have $\to$ has
 - numerous missing periods after formulas (e.g., (10) and before inequality (11))
 - page 19, "This assumption help us ...": help $\to$ helps
 - page 19, "In otherword ...": otherword $\to$ other words
 - page 19, "The through algorithm ...": the sentence is unclear and should be rewritten using academic English
 - page 19, "With the above assumption ...": algorithm $\to$ Algorithm; achieve $\to$ achieves
 - page 19, "The above theorem is almost ...": applies $\to$ apply

## Questions and Comments
1. It would be interesting to see the comparison with recent relevant papers, e.g.,
 - Ogaltsov, Aleksandr, et al. "Adaptive gradient descent for convex and non-convex stochastic optimization." arXiv preprint arXiv:1911.08380 (2019).
 - Défossez, Alexandre, et al. "On the Convergence of Adam and Adagrad." arXiv preprint arXiv:2003.02395 (2020).
2. page 2, "For CIFAR 10 ...", "While for language ...": These claims should be supported by relevant references.
3. page 3, Definition 1: Should be $\mathbb{E}[g(x_k)\mid x_k] = \nabla f(x_k)$ instead of $\mathbb{E}[g(x_k)] = \nabla f(x_k)$. Do you use full expectations in (a) and (b)?
4. page 3, "This is empirically justified ...": To justify this claim, authors should add at least plots comparing $m_k$ and $\sigma_k$ for different optimizers.
5. page 3, Theorem 1: One should explicitly state in the theorem that $f$ is convex.
6. page 4, Assumption 1: Why is $ M$ used to estimate concentration and for bounding $m_k$? It can be the case that concentration is good (e.g., can be achieved via large bathes computed in parallel) while $\max m_k$ is large. Is it possible to get tighter results after the refinement of this assumption? Next, when does this assumption hold for Algorithm 1? It should be stated explicitly in the text with concrete examples of the problems. Why $D^2 = \Omega(M^2)$? The problem should be simpler when $D$ is small, even if $M$ is big. Do you mean $D^2 = O(M^2)$?
7. page 5, Remark 4: it seems you mean standard amplification (the term from complexity theory), not restarts having a different meaning in optimization. One can do that, but this will require additional $O(log|\delta|)$ computations of the functional value of $f$ to choose the best point. For example, it is not possible for the general expectation minimization problem, which restricts the applicability of the results proposed in the paper.
8. page 6, Corollary 5: The proof of this corollary is vital for the paper. Next, the explanation of why this assumption on $M$ and $m_k$ is reasonable should be added.
9. page 6, Corollary 6: The condition $M/m_{avg} \leq T^{1/9}$ should be further explained and motivated. Next, the proof is also vital for the paper.
10. page 7, "The convergence results are ...": This is not true since, in the variance oracle case, authors analyze only non-convex smooth problems and the convergence to the stationary points. It would be interesting to see the analysis for the smooth convex case and see what else we can get with smoothness in the convex case. The authors should explicitly write in the main part of the paper what results they have in the appendix. In the current form, this description is misleading.
11. Experiments: How $m_k$ and $\sigma_k$ were estimated? By definition, these parameters are the full expectation. Do authors apply Monte-Carlo approximation for these parameters in the experiments? Next, there are two idealized and constant baselines in the text (in Sections 3 and F). What baselines did the authors use in the experiments? Have the authors tried all options? If yes, what were the results?
12. page 14, formula for $\hat m_k^2$: Norms are missing for $g_{k-1}, g_{k-2}, \ldots, g_0$.
13. page 14, after the formula for $\hat m_k^2$: independence $\to$ conditional independence.
14. **page 16, inequality above (11):** It is not clear how this inequality was obtained from the above one and the definition of $m$. This part should be clarified.
15. page 16, Remark 9: Why (8) holds in this case? The proof is required.
16. page 16, Remark 9: What does this remark imply? It should be either explicitly stated or removed.
17. **page 18, inequality above (16):** This part should be explained.
18. page 18, application of Markov's inequality: Right-hand side is incorrect.
19. page 19, Assumption 6: Why $D^2 = 4M^2$ is needed? $D$ can be significantly smaller in some cases.
20. **page 21, the second inequality after (18):** It is not clear how this inequality was obtained from the above one and the definition of $m$. This part should be clarified.

## Final remarks

To conclude, the paper focuses on a very important problem but suffers from a number of serious issues described above. Therefore, for me, it is clear that in the current shape, the paper should be rejected. Though there is a chance that the authors will address all my comments, I believe that the paper requires significant improvements and, as a consequence, a new round of reviews.

## AFTER REBUTTAL
First of all, I would like to thank the authors for their detailed response: I realize that the authors spent a lot of effort to address most of my comments and questions. I have read other reviews and the responses by authors. However, I still have questions about the paper, preserving me from increasing the score.

1. **Assumption 1.** Even in the updated form presented in the rebuttal, Assumption 1 is not mathematically rigorous. How can $\mathbb{E}[||g(x,k)||^2]$ be independent of $x$? Can you provide any non-trivial example? Actually, it is highly relevant to my third concern (**Misleading results**) from the weaknesses part: when we use stepsizes dependent on $m_k$ from the future, we implicitly assume that we can use any stepsizes without changing the sequence {$m_k$} (otherwise $m_k$ should depend on $x_k$). Then, if we consider SGD with arbitrary small stepsize, we will get the method that generates arbitrary close points. In these settings, for the majority of problems, the sequence {$m_k$} should be stationary (since we can take arbitrary small stepsize). But it is not stationary, at least in the experiments presented in the paper. **It is a crucial contradiction that significantly decreases the value of the results given in the paper.** In other words, this assumption **never** holds. I understand that the authors simplified the assumption to handle the non-stationarity of the noise. However, the settings considered in the paper are too simplified: they do not cover any problem.
2. **"We could present the paper in terms of $\sigma_k$..."** The paper would significantly benefit from this. Moreover, without resolving the issue mentioned above, it is better to remove the whole part based on the independence of {$m_k$} on $x$.
3. **"Moreover, allowing the step size choice to depend on noise level is a standard approach in almost all SGD analysis."** I agree with the authors. Still, the key difference is that typically it is assumed that **the upper bound** for the noise level is known. It significantly differs from the case when {$m_k$} is known.
4. **"We have compared SGD and Adam in Figure 4 of the appendix (see Appendix A)."** Unfortunately, I have to disagree: the authors presented the behavior of $m_k$ and $\sigma_k$ for SGD and Adam in separate figures -- Figures 1 and 4. There are 10 pages between them, so there is no transparent comparison. It would be much better to see one figure for both algorithms --- this is what I meant by comparison in my review.
5. I still find the proof of Remark 9 to be incomplete: in the proof of the analog of (8) for the case of Remark 9 the authors assumed that $\hat m_{k+1}^2 = \beta \hat m_k^2 + (1-\beta)||g_k||^2$ (in the proof of the last but one inequality), while in this remark it should be $\hat m_{k+1}^p = \beta \hat m_k^p + (1-\beta)||g_k||^p$. Next, what are the assumptions on $p$? Can it be any positive number? If yes, then what is the best choice of $p$? This part requires further discussion and development.
6. **About my 19-th question from the list.** I meant that $D^2$ and $M^2$ could be unrelated in general. It would be interesting to see the discussion on how it influences the convergence rate. The current analysis says that the smaller $D$ is, the better the algorithm converges. It should be discussed how it relates to the empirical findings.
7. The proofs of Corollaries 5, 6, and 7 were given nor in the rebuttal, neither in the paper. Actually, the authors have not updated the paper at all, although many corrections should be applied (at least grammatical errors and misprints noticed above).

To conclude, the most important questions and comments from my review have not been properly addressed by the authors. Therefore, I want to keep my initial score unchanged.

---

> ### Author Response · Authors · 2020-11-23
> **Response to Reviewer 4 part 1**
>
> We appreciate reviewer's detailed feedback, and for acknowledging the problem as important and fundamental. From our understanding, the reviewer's comment centers around the assumption that noise being independent of the algorithm updates, **\bf we kindly refer the reviewer to check our global message discussing different limitation and perspective of our assumption., ([link] https://openreview.net/forum?id=IrofNLZuWF&noteId=cE4TJ_s6-ZK)** In addition, we provide detailed responses to reviewer's concerns below.
>
> > R4: " It is an important assumption and should be stated explicitly in the statements of these theorems. "
>
> We take the criticism on our presentation seriously, as the message has clearly not been parsed successfully, which is our fault. We just want to remark that we have no intention to hide the fact that our assumption is limited, as we have mentioned a non negligible of times that "the noise intensity is decoupled from its location", "$m_k$ is iterate independent", "just of the iteration index $k$", etc. We will take reviewer's suggestion to refine our presentation and make this point clearer in the revision.
>
>
> >R4: "These assumptions are strictly stronger (otherwise, according to Nemirovski and Yudin lower bounds for stochastic non-smooth optimization, it is impossible to get a better rate than SGD has)."
>
> We completely agree with reviewer's comment, what we really mean is that our assumption is a refinement of the standard assumption. We will rephrase the misleading sentence accordingly.
>
> > R4: "why can we choose $\eta_k$ dependent on $\sum_{t=1}^T m_t$ in Corollary 2?... Next, it is unclear how can we know $m_k$ or even $\sum m_t$ to use these stepsizes?"
>
> In section 3, we assume that $m_k$ are given (as they are algorithm and iterate independent). This scenario serves as a strong baseline, both for SGD and adaptive methods, as the parameters are given. In Section 4, we show that when $m_k$ are unknown, adaptive method can still achieve comparable convergence rate. Moreover, allowing the step size choice to depend on noise level is a standard approach in almost all SGD analysis. One way of thinking is that once the step size is selected, it would be optimal for some iteration $T$ (up to rounding error) given the noise levels.
>
> > R4: "Results without the proofs. Some results that play a central role in the paper are left without rigorous proofs and even sketches of the proofs. In particular, the rates from Table 1 are not derived for the noise model considered in Example 1, and the most important results of the paper --- Corollaries 5, 6, and 7 --- are provided without the proofs as well."
>
> The rate in Table 2 can be obtained by simply plug in the parameters $m_k$ in Example 1 to the rates developed in Corollary 2 and Theorem 3. We will detail it in the appendix as suggested by the reviewer, idem for Corollaries 5,6,7.
>
>
> In summary, we thank the reviewer for the critical reviews. We would also appreciate if the reviewer could re-evaluate our contribution after we explained the underlying reasons for the choice of our assumptions. We would be happy to discuss further if reviewer has any other concern or comment.

---

> > ### Author Response · Authors · 2020-11-23
> > **Response to Reviewer 4 part 2**
> >
> > Questions:
> > > 1 "It would be interesting to see the comparison with recent relevant papers"
> >
> > We will add the references mentioned by the reviewer. As in the standard uniform bound setting, SGD is optimal, the mentioned papers do not show non-trivial improvement in the convergence rate.
> >
> > > 2 "For CIFAR 10 ...", "While for language ...": These claims should be supported by relevant references.“
> >
> > These choices are commonly used in practice (not even just the optimization algorithm but also the learning rate scheduling) We will add reference to support our claim. ({\color{red} Add reference?})
> >
> > > 3 "Definition 1: Should be $\mathbb{E}[g(x_k) | x_k] = \nabla f(x_k)$ instead of $\mathbb{E}[g(x_k)] = \nabla f(x_k)$."
> >
> > We agree with the reviewer's comment, it is $\mathbb{E}[g(x_k) | x_k] = \nabla f(x_k)$.
> >
> > > 4 "This is empirically justified ...: To justify this claim, authors should add at least plots comparing $m_k$ and $\sigma_k$  for different optimizers."
> >
> > We have compared SGD and Adam in Figure 4 of appendix(see Appendix A). We will include a more detailed comparison in the revision by allowing different learning rate for a given algorithm. In the LSTM task, the shape in are quite similar for various algorithms and stepsizes; In CIFAR-10, the shape varies more significantly. This empirical observation justifies that our assumption is not completely out of range.
> >
> > > 5 "Theorem 1: One should explicitly state in the theorem that  is convex."
> >
> > We have mentioned in text that "Let f be convex and differentiable" six lines above Theorem 1.
> >
> > > 6 "Assumption 1: Why is $M$ used to estimate concentration and for bounding $m_k$? It can be the case that concentration is good (e.g., can be achieved via large bathes computed in parallel) while $\max m_k$ is large. Is it possible to get tighter results after the refinement of this assumption? Next, when does this assumption hold for Algorithm 1? It should be stated explicitly in the text with concrete examples of the problems. Why $D^2 = \Omega(M^2)$? The problem should be simpler when $D^2$ is small."
> >
> > We thank reviewer's remark. Our proof holds indifferently when $D^2$ is unrelated to $M^2$, i.e. separating the roles of $D^2$ (concentration) and $M^2$ (absolute norm). The reason we make such assumption is to facilitate comparison, as the convergence rate involves different accumulations of $m_k$ or $\sigma_k$ and can not be easily simplified to a form $O(T^\beta)$ in general. Even under this simplification, the comparison is not  straightforward.
> >
> > > 7 "page 5, Remark 4: it seems you mean standard amplification (the term from complexity theory), not restarts having a different meaning in optimization. One can do that, but this will require additional $O(\log(\delta))$ computations of the functional value of $f$ to choose the best point. For example, it is not possible for the general expectation minimization problem, which restricts the applicability of the results proposed in the paper."
> >
> > We thank the reviewer for mentioning the appropriate term. We will replaced the phrasing as "amplification by repeated trials" following reviwer's suggestion.
> >
> > > 8 and 9 "The proof of the corollaries are vital for the paper. Next, the explanation of why this assumption on $M$ and $m_k$ is reasonable should be added."
> >
> > The proof is a simple plug in of the assumption, we will make it clearer in the revision. The convergence rate in the general setting depends on the convergence of different power series of $m_k$. The case when $M/ \min m_k$ is of constant level is not interesting, as it is essentially a stationary noise. The interesting case would be $M/ \min m_k$ is a polynomial of $T$. We do not aim to say that the specific choice $T^{1/9}$ is the only possible setting, instead what we are trying to say is that as soon as this condition holds, then the adaptive method converges in the same order as the idealized baseline. This gives a rough idea how weak/strong our convergence result stands compared to the idealized setting.
> >
> > > 10 page 7, "The convergence results are ...": This is not true since, in the variance oracle case, authors analyze only non-convex smooth problems and the convergence to the stationary points. It would be interesting to see the analysis for the smooth convex case and see what else we can get with smoothness in the convex case. The authors should explicitly write in the main part of the paper what results they have in the appendix. In the current form, this description is misleading.
> >
> > The convergence proof under the variance oracle is essentially the same as the moment oracle. We have omitted the proof to avoid too much redundancy. Moreover, the non-convex case also provide strong evidence that the proof can be followed similarly in the convex setting. We will provide a sketch of the proof in the appendix of revision.

---

> > > ### Author Response · Authors · 2020-11-23
> > > **Response to Reviewer 4 Part 3**
> > >
> > > > 11 Experiments: How $m_k$ and $\sigma_k$ were estimated? By definition, these parameters are the full expectation. Do authors apply Monte-Carlo approximation for these parameters in the experiments? Next, there are two idealized and constant baselines in the text (in Sections 3 and F). What baselines did the authors use in the experiments? Have the authors tried all options? If yes, what were the results?
> > >
> > > We use the additive noise model in our experiment which allows us to impose $\sigma_k$. Hence we apply the baselines based on variance oracle.
> > >
> > > > 12, 13: we thank reviewer for pointing out the in-rigorous and typo.
> > >
> > > > 14 **page 16, inequality above (11)**
> > >
> > > This is indeed less straightforward, the details are as follows.
> > > \begin{align*}
> > >      \frac{1}{2m^3} \sum_{k=1}^T  {(\hat{m}_k-m_k)_+^2  }  \le &
> > >     \frac{1}{2m^3} \cdot 8(D^2 + M^2 ) T^{2/3} \ln (T^{2/3}) \\\\
> > > = & \frac{4}{m^3} \cdot (D^2 + M^2 ) T^{2/3} \frac{2}{3} \ln (T)\\\\
> > > = & \frac{8}{3m^3} \cdot (D^2 + M^2 ) T^{2/3} \ln (T)
> > > \end{align*}
> > > By definition, $m = 4\sqrt{D^2+M^2} T^{-1/9} (\ln (T))^{1/2}$, we have $m^2 = 16 (D^2+M^2)T^{-2/9} \ln (T)$. Hence
> > > $$ \frac{8}{3m^3} \cdot (D^2 + M^2 ) T^{2/3} \ln (T) = \frac{1}{6m} \cdot  T^{8/9} $$
> > > Finally, we show that
> > > $$\frac{1}{6m} T^{8/9} \le \frac{T}{4(M+m)},$$
> > > which rewrite as $2M +2m \le 3m T^{1/9}$. On one hand $T^{1/9} \ge 1$ hence $2m T^{1/9} \ge 2m $. On the other hand $m T^{1/9} =  4\sqrt{D^2+M^2} (\ln (T))^{\frac{1}{2}} \ge 4M$ when $T \ge 3$. Hence the desired inequality holds.
> > >
> > > > 15 "page 16, Remark 9: Why (8) holds in this case? The proof is required."
> > >
> > > The proof follows exactly as (8).
> > > \begin{align*}
> > > \sum_{k=1}^{T} \mathbb{E}[\eta_k^2 m_k^2] & = c^2\sum_{k=1}^{T} \mathbb{E}\left [\frac{m_k^2}{(\hat{m}_k^p + m^p)^{2/p}} \right ]  \\\\\\
> > > & =  c^2 \sum \mathbb{E}\left [\frac{m_k^2 - \hat{m}_k^2}{(\hat{m}_k^p + m^p)^{2/p}} \right ] + \sum \mathbb{E}\left [\frac{ \hat{m}_k^2 }{(\hat{m}_k^p + m^p)^{2/p}} \right ]  \\\\
> > > & \le c^2 \left ( \frac{1}{m^2} \sum \mathbb{E}\left [ | m_k^2 - \hat{m}_k^2| \right ] + T  \right ) \\\\
> > > & \le c^2 \left ( \frac{(M^2+D^2) T^{2/3}\ln(T^{2/3})}{m^2} + T  \right )  \le 3 c^2T
> > > \end{align*}
> > >
> > >
> > >
> > > > 16 "page 16, Remark 9: What does this remark imply? It should be either explicitly stated or removed."
> > >
> > > This remark imply that we can use the stepsize of type
> > > $$ \hat{m}_{k+1}^p = \beta \hat{m}_k^p + (1-\beta) |g_k|^p,$$
> > > which is the variant mentioned in paragraph "**Variants on stepsize**" on page 7
> > >
> > > > 17 "page 18, inequality above (16): This part should be explained."
> > >
> > > Idem to the explanation in 15.
> > >
> > > > 18 "page 18, application of Markov's inequality: Right-hand side is incorrect."
> > >
> > > Thank you for pointing out. The typo is indeed on the left hand side, it should be with probability at least $3/4$
> > > $$\sum  {(\hat{m}_k-\lambda_k)_+ } \le 4\mathbb{E} [\sum  {|\hat{m}_k-\lambda_k| }]. $$
> > >
> > > > 19 "page 19, Assumption 6: Why $D^2 = 4M^2$ is needed?  can be significantly smaller in some cases."
> > >
> > > This is an igorance of our part, it should be $D^2= \Omega(M^2)$. The case $D^2 = 4M^2$ is the specific case for Example 1.
> > >
> > > >20 "page 21, the second inequality after (18): It is not clear how this inequality was obtained from the above one and the definition of . This part should be clarified."
> > >
> > > Idem to 15 and 17.

---

### Official Review · AnonReviewer2 · 2020-10-28
**Nice work on stepsize selection under non-adaptive noise. A few things can be improved.**

**Rating:** 5
**Confidence:** 4

**Review:**

This paper studies gradient-based stochastic optimization algorithms which incorporate (estimates of) the noise statistics in the adaptive stepsize design. Starting from the standard analysis of SGD with adaptive steps (Thm 1) the authors show in Cor 1 how, using a second-moment-dependent learning rate, one can “accelerate” (see comment later) the convergence of SGD. Next, the authors show (Thm 3) that one can recover a similar result by estimating the second moment using an exponential moving average.

The paper is well written, and I think Thm3 is novel, correct and interesting. However, I think the authors should address the following points to improve their work and make the paper publishable:

1) The authors should discuss the performance of their method on neural nets more scientifically: the architecture and the hyperparameters selection of Figure 3 are not discussed. Also, just one architecture/dataset is presented. @authors, also how did you select the parameter beta in the experiments? Can you show us more results maybe on deeper models?

2) We all know very well that the standard analysis of non-convex gradient descent (which the authors adapt here) does not lead to tight upper bounds. Hence, it would instead be interesting to also study the behavior of the methods proposed by the authors on convex problems and to provide some tight rates backed up on well-controlled experiments on ill-conditioned linear and logistic regression. This would better illustrate the idea. In short, since this phenomenon is also present in convex problems, I think the authors should provide a discussion on this too.

3) The paper focuses a bit too much on example 1, showing very fast rates. It would be great to also compute the speed-up predicted by the algorithm based on the actual variance of a real-world optimization problem. How big is this theoretical speed-up given a real-world variance?

If the authors provide a good answer and are able to provide a convincing real-world example for point (3) then I will raise my score to a weak accept.

---

> ### Author Response · Authors · 2020-11-23
> **Response to Reviewer 2**
>
> We thank the reviewer for considering our work as novel and interesting. We provide detailed responses to reviewer's concerns below.
>
> > R2:  "Discuss the performance of their method on neural nets more scientifically"
>
> The hyperparameters, architecture and dataset details are included in Appendix A. The implementation remains largely unmodified, and anything not mentioned is set the same way as the original codebase we referred. We also included the code of our LSTM experiments for reproducibility. Our algorithm in Figure 3 is tested on two dataset Cifar10 and PTB using different models.
>
> > R2  "It would instead be interesting to also study the behavior of the methods proposed by the authors on convex problems...Since this phenomenon is also present in convex problems, I think the authors should provide a discussion on this too."
>
> We thank the reviewer for this suggestion. **Our main results presented in Theorem 1 and 3 are indeed for convex setting.** Studying whether adaptivity can improve dependency on condition number is an challenging and interesting question, which would depend on how large can the noise changes. This can be an interesting future problem.
>
> > R2 "It would be great to also compute the speed-up predicted by the algorithm based on the actual variance of a real-world optimization problem."
>
> We thank reviewer's suggestion. The speed up is on the order of $T^{1/9}$, which usually gives vacuous improvement given the existing constants. However, we hope to point out that our work is the first to identify a theoretical gap between SGD and adaptive methods in the non-stationary setting.

---

> > ### Comment · AnonReviewer2 · 2020-11-24
> > **Think the paper will benefit from some additional experiments**
> >
> > Thanks a lot for the reply, and sorry I did not catch that there was a convexity assumption (please highlight it better). Since the authors focus on convex settings, I think they should study/present the performance on real-world (non synthetic) convex examples before jumping to neural nets: logistic regression, hinge loss, etc.

---

### Official Review · AnonReviewer3 · 2020-10-28
**Time-dependent stochastic gradient moment assumptions are artificial**

**Rating:** 4
**Confidence:** 4

**Review:**

This paper studies the problem of stochastic optimization where the gradient noise process is non-stationary. Based on a general convergence results based on a general sequence for the second moments of the stochastic gradient norms and a general stepsize sequence, the authors propose to use an online estimation procedure for the gradient norm second moments, in order to mimic the behavior of the ``idealized'' stepsize sequence. Finite-time convergence rates are established for the algorithms with adaptive stepsize, leading to an acceleration effect in certain regimes for the non-stationarity.

Both the adaptive stochastic gradient algorithms and handling non-stationary noise are indeed interesting problems. However, the two problems are of different natures and this paper attempts to address one of them from the standing point of another, which is not particularly convincing. Indeed, for any stochastic optimization problem for supervised learning (including training neural nets that the authors mentioned), the stochastic gradient oracles are i.i.d. copies of stochastic gradient function $\nabla f (\theta; \xi_i)$, where $\theta$ is the model parameter and $\xi_i$ are i.i.d. data points. The sequence of functions $(\nabla f (\cdot; \xi_i))_{i = 1}^{+\infty}$ are not only stationary but also i.i.d., and the only reason for observing difference levels of noise variances is the point at which the stochastic gradient is evaluated. In the introductory parts, authors realized the existing works that assume bounds on the noise variance based on the norm of current iterates. Though the noise variance does not always scale like this in neural network training, it is due to more complicated landscape and noise structure, but not due to the time steps.

Therefore, it is quite artificial to modle the noise process as a non-stationary sequence that varies with time. Indeed, if you use a difference algorithm, it is reasonable to expect that the sequence of points at which the stochastic gradients oracle are evaluated can change a lot, leading to a totally different noise variance sequence. There are applications, such as real-time stochastic control with a non-stationary environment, where the noises are intrinsically changing with time. But it is not the case with classical supervised learning.

On a related note, under the stochastic gradient moment sequence in Example 1 in the paper, it appears that the best thing to do might be to only use the first 1/5 and last 1/5 proportion of the data stream, while doing nothing with the middle 3/5 of the data when the noise is large. Up to a constant factor of 5, this naive algorithm gives the convergence rate the same as in the model where the noise is always of order $T^{-\alpha}$, which is better than the results under ``idealized'' stepsize sequence predicted by the paper. In training neural networks, however, this will become a stupid algorithm, which does nothing but wasting 3/5 of the data. When it comes back to updating the parameter using the last 1/5 of the data, the noise will be still large because the iterates enters a region of large stochastic gradients. If this naive algorithm can be challenged by this reasoning, how can we argue that the algorithms designed in the paper does not suffer from this?

Besides, the quantity $m_k^2$ is defined as the second moment for the stochastic gradient at the $k$-th iterate. It is larger than the squared norm of the population-level gradient, by Cauchy-Schwartz inequality. Imposing an upper bound on this quantity that depends on $T$ is implicitly assuming some convergence property of the algorithm. And this bound is $T{-\alpha}$ at the initial point, which means that the algorithm starts at a near-stationary point. This makes the theory even more artificial. Even in an actual real-time non-stationary environment, it is ok to assume a sequence of noise variances, but it is not convincing to assume a bound on the gradient itself.

The adaptive stochastic gradient algorithm presented in this paper is interesting, and may make an interesting step towards more understanding into adaptive SGD methods based on online moment estimates. I would encourage the authors to dig deeper in this direction under a suitable set of assumptions, and show some real acceleration effect.

---

> ### Author Response · Authors · 2020-11-23
> **Response to Review 3 part 1**
>
> We thank the reviewer for the critical reviews and we appreciate reviewer for considering the problem we study as important. From our understanding, the reviewer's comment centers around the assumption that noise being independent of the algorithm updates, **we kindly refer the reviewer to check our global message discussing different limitation and perspective of our assumption**,([link] https://openreview.net/forum?id=IrofNLZuWF&noteId=cE4TJ_s6-ZK). In addition, we provide detailed responses to reviewer's concerns below.
>
> >R3: “It is quite artificial to model the noise process as a non-stationary sequence that varies with time. ”
>
> First, we do assume that the stochastic oracle is **iterate and algorithm independent**, and, **we admit that our assumption has limitations**. Our intention is to study how a static non-stationary noise influence the convergence of different algorithms. (static in the sense algorithm independent) Even though simplified, this is a new and interesting scenario, as the empirical observation shows that the change in second moment/variance is huge, up to a ratio as large as $10^6$, in deep learning tasks. Imposing a non-stationarity assumption is thus necessary because the standard uniform bound assumption will lead to very pessimistic convergence analysis, unable to reflect the reality in this regime. This is why we introduce such refinement of the standard hypothesis.
>
> Second, we have empirically compared the noise obtained by SGD and Adam in Figure 4 of appendix(see Appendix A). We will include a more detailed comparison in the revision by allowing different learning rate for a given algorithm. In the LSTM task, the shape in are quite similar for various algorithms and stepsizes; In CIFAR-10, the shape varies more significantly. This empirical observation justifies that our assumption is not completely out of range.
>
> Third,  we hope to point out that the condition described by the researcher where the noise level is a function of the state variable cannot be analyzed unless one could bound the noise by the amount of descent (gradient norm, suboptimality, distance to optimal, etc). This naturally leaves out a class of interesting functions. For general noise distribution as a function of the variable x, one can easily construct examples such that no algorithm can exploit the noise structure. Hence, our setting simplifies the problem to the extent where one could get some interesting results.
>
> Finally, regardless of whether this assumption is oversimplified or not, we believe our result is full of interest as it is the first result showing that the adaptive methods can theoretically outperform SGD, depending on how the second moment/variance changes. Even in such setting, it is completely non-trivial to derive the convergence analysis when that the parameters $m_k$ or $\sigma_k$ are not given.
>
> >R3: "It appears that the best thing to do might be to only use the first 1/5 and last 1/5 proportion of the data stream, while doing nothing with the middle 3/5 of the data when the noise is large.  If this naive algorithm can be challenged by this reasoning, how can we argue that the algorithms designed in the paper does not suffer from this?"
>
> We thanks the reviewer for pointing out such interesting thought. First, when using the algorithm proposed by the reviewer (only use the first 1/5 and last 1/5 proportion), the convergence rate would be $O(T^{\frac{1}{2} + \alpha})$ which is the same as the ``idealized'' stepsize sequence. However, this algorithm won't perform well when the shape of the noise is in a different form (for example the first 1/5 and last 1/5 proportion equals 1 and the middle part is small). **In other words, the design of such algorithm relies on the knowing the shape of $m_k$ or $\sigma_k$.** The power of the adaptive method (with moment estimation) is that it does not need to know the shape in advance, and still achieve a comparable rate as if the parameters are known, whereas the naive algorithm mentioned requires knowing the noise shape in advance.
>
> The reviewer's criticism on the practical implication is perfectly valid. As we mentioned in our general comment, the static non-stationary noise model is clearly a simplification of the real cases. Even though simplified, we believe it is one step further than the standard uniform bound assumption.

---

> > ### Author Response · Authors · 2020-11-23
> > **Response to Review 3 part 2 (need to cut due to characters limitation)**
> >
> > >R3: "Imposing an upper bound on this quantity that depends on  is implicitly assuming some convergence property of the algorithm..."
> >
> > We agree with the reviewer that bounding the second moment implicitly bounds the gradient norm (from Cauchy-Schwartz), i.e. imposing a $T^{-\alpha}$ type bound directly implies that we are closed to a stationary point. This phenomenon won't happen when we consider the variance case $\sigma_k$.  Moreover, we remark that our result is scale invariant. In other words, the value should not be considered literally, as multiplying $m_k$ or $\sigma_k$ by an arbitrary constant does not change the conclusion. **What it really matters is the ratio between the maximum $m_k$ and the minimum $m_k$**. Moreover, if we look at the empirical noise obtained in the Transformer/LSTM training, the smallest gradient norm/ variance is achieved around the initialization, which coincides with the assumption.
> >
> > In summary, we thank the reviewer for the critical reviews. We would also appreciate if the reviewer could re-evaluate our contribution after we explained the underlying reasons for the choice of our assumptions. We would be happy to discuss further if reviewer has any other concern or comment.

---

### Official Review · AnonReviewer1 · 2020-10-29
**Stochastic Approximation with  non-stationary noise**

**Rating:** 3
**Confidence:** 5

**Review:**

The objective of the paper is to provide a theoretical justification for the value of using adaptive learning steps. The paper presents two results. The first is essentially of theoretical interest, and assumes that the noise level indicators defined in Eq. (1) [but which are difficult to understand at this level of the paper] are known. The second is more practical: it shows that a variant of the RMSprop algorithm achieves the same results as the "theoretical" algorithm.

I perfectly understand the definition of the second moment $m^2(x)$ and $\sigma^2(x)$. These quantities depend on the current value of the parameter. This is a feature of the function we are trying to optimize, and this is perfectly clear. The Definition 1 of the non-stationary noise oracle is more far-fetched: "The stochasticity of the problem is  governed by a a sequence of second moment $\{m_k\}$ and variance $\sigma_k^2$". The definition is not even very well written and explained. This is an unconditional exceptation: it means in particular that the expectation is taken wrt to the distribution of the initial value  of the parameter (I guess therefore that the expectation is also taken on the initial condition, which can be concentrated on a point).  Even if we take a  this is very difficult to check, except in the situation in which the "noisy gradient" used in the procedure is the true gradient affected by some additive noise, independent from the current value of the parameter. The authors feel this embrassment in the sentence that their "goal is to demystify the correlation between the noise intensity and the performance of the algorithm" and two lines later "the noise intensity" (what does it mean "intensity" ?) is "decoupled" from the location, in strict contradiction of their earlier definiton of second moment and variance which are, of course, location dependent. We can of course argue that in most analysis the variance (or even the conditional variance) of the gradient is bounded, so that this may appear as a relaxation of the previous assumptions... Theorem 1 is not surprising, same three-lines proof where the bound on the gradient noise variance is simply replaced by a time-dependent variance. They illustrate the result with a very artificial example (Example 1), which substantiate an earier claim (that the adaptive stepsize can achieve a faster rate of convergence by a factor which is polynomial in T). In section 4, the authors start to consider the more interesting "adaptive" scenario. The algorithm is a simplified version of RMSprop in which a "global" scale is applied to the gradient instead of an "component-dependent" factor applied  on each individual component of the gradient.  The authors formulate Assumption 1, which a bit strange because the assumption is algorithm dependent (the unconditional variance of the algorithm after $k$-step depends on the algorithm itself, so we have to guess that Assumption 1 is formulated for the adaptive SGD algorithm... but then the quantities depend upon the constant $m$  and $c$ which are later optimized in the theorem. Everything is written as if the noise variance depends only on the iteration index, independently of the algorithm itself... Given this remark, it is very difficult to understand what theorem 3 is about... This really gives the impression of a snake biting its tail, because the definition of $m_k$ depends on the law of the  iterate and therefore on the SA algorithm, which we then allow ourselves to optimize the parameters of the algorithm but without affecting the law of iteration (because the m_k are always the same). Strange... Of course, the only scenario where this makes sense is when $g_k= \nabla f(x_k) + \sigma^_k Z_k$, i.e. when the gradient noise depends only on the iteration index but not on the current  value of the iterate.
 I find the paper interesting, but the conclusions must be rewritten very clearly so as not to mislead the reader. You are in fact studying the stochastic approximation with an error that would have a variance profile, perhaps unknown, and depending on the iteration index but not on the law of iterations. As a result, the conclusions have much less force.

---

> ### Author Response · Authors · 2020-11-23
> **Response to Review 1**
>
> We thank the reviewer for the critical reviews and we appreciate reviewer for considering the problem we study as important. From our understanding, the reviewer's comment centers around the assumption that noise being independent of the algorithm updates,  **we kindly refer the reviewer to check our global message discussing different limitation and perspective of our assumption**, ([link] https://openreview.net/forum?id=IrofNLZuWF&noteId=cE4TJ_s6-ZK). In addition, we provide detailed responses to reviewer's concerns below.
>
> First, we do assume that the stochastic oracle is **iterate and algorithm independent**, and, **we admit that our assumption has limitations**. We agree that this is a novel and non-standard setting that we should emphasize further in the paper. We would be happy to further discuss on it. Meanwhile, we take the criticism on our presentation seriously, as the message has clearly not been parsed successfully, which is our fault. We just want to remark that we have no intention to hide the fact that our assumption is limited, as we have mentioned a non negligible of times that "the noise intensity is decoupled from its location", "$m_k$ is iterate independent", "just of the iteration index $k$", etc.
>
> Second, we have empirically compared the noise obtained by SGD and Adam in Figure 4 of appendix(see Appendix A). We will include a more detailed comparison in the revision by allowing different learning rate for a given algorithm. In the LSTM task, the shape in are quite similar for various algorithms and stepsizes; In CIFAR-10, the shape varies more significantly. This empirical observation justifies that our assumption is not completely out of range.
>
> Third,  we hope to point out that the condition described by the reviewer where the noise level is a function of the state variable cannot be analyzed unless one could bound the noise by the amount of descent (gradient norm, suboptimality, distance to optimal, etc). This naturally leaves out a class of interesting functions. For general noise distribution as a function of the variable x, one can easily construct examples such that no algorithm can exploit the noise structure. Hence, our setting simplifies the problem to the extent where one could get some interesting results.
>
> Finally, regardless of whether this assumption is oversimplified or not, we believe our result is full of interest as it is the first result showing that the adaptive methods can theoretically outperform SGD, depending on how the second moment/variance changes. Even in such setting, it is completely non-trivial to derive the convergence analysis when that the parameters $m_k$ or $\sigma_k$ are not given.  We believe our work is a good first step towards understanding the behavior of algorithms under non-stationary noise.
>
> In summary, we thank the reviewer for the critical reviews. We would also appreciate if the reviewer could re-evaluate our contribution after we explained the underlying reasons for the choice of our assumptions. We would be happy to discuss further if reviewer has any other concern or comment.

---

### Author Response · Authors · 2020-11-23
**Addressing the common concerns on Non-stationary Assumption**

We thank the valuable feedback from all the reviewers. We would like to address a common concern raised by the reviewers (R1,R3,R4) regarding the lack of clarity on the non-stationary assumption. In short,  we do assume that the stochastic oracle is **iterate and algorithm independent**, and, **we admit that our assumption has limitations**:

**Assumption 1**: We assume existence of sequences of absolute constants $(m_k)_{k \in \mathbb{N}}$ or $(\sigma_k)_{k \in \mathbb{N}}$, which does not dependent on algorithm and iterates, such that the stochastic gradient oracle takes as input any point $x \in \mathbb{R}^n$ and any iteration number $k \in \mathbb{N}$, outputs a random vector $g(x,k)$ such that $\mathbb{E}[g(x,k)] = \nabla f(x)$, where the expectation is taken with respect to the randomization of the gradient oracle, and either
1. $\mathbb{E}[ \| g(x,k)\|^2] = m_k^2$ **for any $ x  \in \mathbb{R}^n$**;
2. $\mathbb{E}[ \| g(x,k)- \nabla f(x) \|^2] = \sigma_k^2$ **for any $x  \in \mathbb{R}^n$**.

In other words, the gradient variance on the 10-th iteration of SGD is assumed to be the same as the gradient variance on the 10-th iteration of Adam.

We take the criticism on our presentation seriously, as the message has clearly not been parsed successfully, which is our fault. We just want to remark that we have no intention to hide the fact that our assumption is limited, as we have mentioned a non negligible of times that "the noise intensity is decoupled from its location", "$m_k$ is
iterate independent", "just of the iteration index $k$", etc. We will take reviewer's suggestion to refine our presentation and make this point clearer in the revision.

**Regarding the iterate independence assumption.** We completely agree with the reviewers that this assumption is arguable, especially the one on the second moment. In fact, we have clearly stated in page 7 of the main paper that

 >"However, there is some unnaturalness underlying the non-stationary oracle on $m_k$. Indeed, it is hard to argue that the second moment is iterate independent since $\mathbb{E}[ \| g(x,k)\|^2] = \| \nabla f(x_k) \|^2 + \mathbb{E}[ \| g(x,k) - \nabla f(x_k) \|^2]$. Even though the influence of $\| \nabla f(x_k) \|^2$ might be minor when the variance is high (e.g. as in Figure 1), it is still changing the second moment."

While as the second moment oracle is less natural, the variance oracle is theoretically sound, for example when the noise is additive, i.e.  $g(x,k) \sim \nabla f(x_k) + \mathcal{N}(0,\sigma_k^2)$. Our result holds indifferently for the variance oracle. We could present the paper in terms of $\sigma_k$ if reviewers find that would be more appropriate.

**Regarding the algorithm independence assumption.** We also agree with the reviewers that in the most general setting, the noise would be algorithm dependent, as different algorithm simply arrive in different locations. We agree that investigating how the noise changes according to different algorithm is a very interesting direction. However, this is highly non-trivial even in the simplest quadratic cases, as it involves combination of data distribution, model architecture, etc.

Our intention is to study how a static non-stationary noise influence the convergence of different algorithms. (static in the sense algorithm independent) Even though simplified, this is a new and interesting scenario, as the empirical observation shows that the change in second moment/variance is huge, up to a ratio as large as $10^6$, in deep learning tasks. Imposing a non-stationarity assumption is thus necessary because the standard uniform bound assumption will lead to very pessimistic convergence analysis, unable to reflect the reality in this regime. This is why we introduce such refinement of the standard hypothesis. **While as the main criticisms are regarding the over simplification and unnaturalness of our assumption, our main contribution has been largely dismissed by the reviewers.  Regardless of whether or not this assumption is oversimplified, we believe our result is full of interest as it clearly illustrates the contrast between SGD and adaptive methods under the considered setting. Though it is a simplified setting, this is the first result showing that the adaptive methods can theoretically outperform SGD, which provides new perspective on the success of adaptive methods.**

---

### Decision · Program_Chairs · 2021-01-07
**Final Decision**

**Decision:**

Reject

**Comment:**

The paper studies the problem of stochastic optimization where the gradient noise process is non-stationary. While this is an important problem in the community, the reviewers find that the assumptions are poorly justified. While the authors provided extensive feedback, the reviewers did not change their initial assessment. This paper can therefore not be accepted in its current form. I think the reviewers provided some very critical and useful feedback and I therefore strongly encourage the authors to take advantage of this feedback to resubmit their paper to another venue.